METHODS AND RESOURCES

# An update to Hippocampome.org by integrating single-cell phenotypes with circuit function in vivo

**Alberto Sanchez-Aguilera**[1], **Diek W. Wheeler**[2], **Teresa Jurado-Parras**[1],
**Manuel Valero**[1,3], **Miriam S. Nokia**[1,4,5], **Elena Cid**[1], **Ivan Fernandez-Lamo**[1], **Nate Sutton**[2],
**Daniel García-Rincón**[1], **Liset M. de la Prida**[1]*, **Giorgio A. Ascoli**[2]*

1 Instituto Cajal CSIC, Madrid, Spain, 2 Bioengineering Department, Volgenau School of Engineering,
George Mason University, Virginia, United States of America, 3 NYU Neuroscience Institute, New York,
United States of America, 4 Department of Psychology, University of Jyvaskyla, Jyvaskyla, Finland,
5 Neuroscience Center, HiLIFE, University of Helsinki, Helsinki, Finland

☯ These authors contributed equally to this work.
* lmprida@cajal.csic.es (LMP); ascoli@gmu.edu (GAA)

University of Vienna, AUSTRIA

**Data Availability Statement:** All relevant data are
within the paper and its Supporting Information
files or are found at http://www.hippocampome.
org.

## Abstract

Understanding brain operation demands linking basic behavioral traits to cell-type specific
dynamics of different brain-wide subcircuits. This requires a system to classify the basic
operational modes of neurons and circuits. Single-cell phenotyping of firing behavior during
ongoing oscillations in vivo has provided a large body of evidence on entorhinal–hippocam-
pal function, but data are dispersed and diverse. Here, we mined literature to search for
information regarding the phase-timing dynamics of over 100 hippocampal/entorhinal neu-
ron types defined in Hippocampome.org. We identified missing and unresolved pieces of
knowledge (e.g., the preferred theta phase for a specific neuron type) and complemented
the dataset with our own new data. By confronting the effect of brain state and recording
methods, we highlight the equivalences and differences across conditions and offer a num-
ber of novel observations. We show how a heuristic approach based on oscillatory features
of morphologically identified neurons can aid in classifying extracellular recordings of single
cells and discuss future opportunities and challenges towards integrating single-cell pheno-
types with circuit function.

## Introduction

Understanding brain function requires an integrative approach interrelating neurons, circuits,
and behaviors. In linking through organization levels, we need to assimilate information about
cell types and their intrinsic properties with knowledge about connectivity patterns and behav-
ioral influences. Therefore, scaling from single to multiple cells is critical. However, being pre-
cise about neuronal identity, as defined by morphology and neurochemical markers, entails
recording and labeling neurons with glass pipettes [1], which is limited to very few cells in vivo
[2]. While recording hundreds of cells extracellularly provides an alternative, this approach

**Funding:** The authors received funding from the following sources: 1: National Institutes of Health (https://www.nih.gov/grants-funding) (R01NS39600 and U01MH114829) [GAA] 2: Spanish Ministerio de Economía y Competitividad (MINECO) (https://portal.mineco.gob.es/en-us/Pages/default.aspx) (RTI2018-098581-B-I00 and FJCI-2017-32719) [LMP and ASA] 3: Academy of Finland (https://www.aka.fi/en/) (grant nr. 275954) [MSN] 4: European Molecular Biology Organization (https://www.embo.org/funding-awards/fellowships) (EMBO ALTF 1161-2017) [MV] 5: Human Frontiers Science Program (https://www.hfsp.org/funding/hfsp-funding/) (LT0000717/2018) [MV] The funders had no role in study design, data collection and analysis, decision to publish, or preparation of the manuscript.

**Competing interests:** The authors have declared that no competing interests exist.

**Abbreviations:** AHP, afterhyperpolarization; CCK, cholecystokinin; DG, dentate gyrus; DS, dentate spikes; FBS, fetal bovine serum; LED, light-emitting diode; LFP, local field potential; MEC, medial entorhinal cortex; MVL, mean vector length; NPY, neuropeptide-y; O-LM, oriens-lacunosum moleculare; PBS, phosphate buffered saline; PV, parvalbumin; PV-BC, PV basket cells; RT, room temperature; SOM, somatostatin; SP, stratum pyramidale; Sub, subiculum; SWR, sharp-wave ripple; TSWB, transient slow-wave bursting.

remains relatively blind to the taxonomy of cell diversity that is essential to elucidate microcircuits [3–5]. Heterogeneity and a lack of precise genetic tools to map cell types unambiguously complicate current approaches, especially for higher-order cortical regions, where functional mapping is more elusive.

The discovery that the diversity of hippocampal glutamatergic and GABAergic cells can be better understood by incorporating information about their firing dynamics with respect to ongoing oscillations changed the game [6]. Single-cell phenotyping based on firing behavior in vivo has provided a large body of evidence to characterize microcircuit operation [7–13]. This approach can be also used to inform blind extracellular recordings. For instance, the characteristic theta phase preference and sharp-wave ripple (SWR) firing of parvalbumin (PV) and cholecystokinin (CCK) CA1 basket cells, together with their gamma entrainment, facilitate identification with multielectrode and/or tetrode recordings [14,15]. This, together with promoter-specific optogenetic tagging of cells, allows for functional mapping of basic microcircuits [16–18], which emerge as the mesoscopic building blocks of elementary cognitive functions [19]. Yet, in many cases, genetic promoters do not fully guarantee selectivity.

Assimilating all the knowledge about hippocampal and entorhinal cells in a common neuron classification framework would permit the annotation of high-throughput recordings and provide new opportunities for crucially integrating intrinsic single-cell phenotyping with microcircuit function. Although in vivo approaches are the goal, the community efforts to build a systematic cellular taxonomy integrating a wealth of transcriptomic, morphological, and electrophysiological single-cell data rely on in vitro strategies [20]. Even under this controlled condition, phenotypic variations in the morphoelectrical space are largely independent of transcriptomic variability within specific cell types, suggesting that the concept of cellular identity is more complex than originally thought [21]. Possibly, functional influences, such as maturation, learning and plasticity, adherence to different engrams, and/or specific cognitive/behavioral correlates, all should be incorporated into cell-type definition. Unfortunately, single-cell validated electrophysiological in vivo data are dispersed and involve different recording conditions, species, and methodologies.

Here, we mined literature to search for and carefully annotate information regarding the phase-timing dynamics of single cells identified and labeled with various in vivo recording techniques. We then associated these data with specific neuron types from Hippocampome.org, a mature knowledge base that defines more than a hundred hippocampal and entorhinal cell types classified according to the main neurotransmitter, axonal and dendritic patterns, synaptic specificity, in vitro electrophysiology, and molecular biomarkers [22]. Moreover, we complemented this newly organized evidence with our own new recordings from underrepresented cell types. The synergistic collection of novel experimental measurements in parallel with our quantification of literature-derived evidence guided a systematic process of data normalization that was crucial for subsequent analysis. By integrating in vivo functional pieces of knowledge (e.g., preferred theta phases and SWR-associated firing dynamics for specific neuron types) into Hippocampome.org, we extended the previous classification of certain neuron types using novel information on oscillatory dynamics [22]. Furthermore, we confronted the effects of experimental method (i.e., freely moving versus head-fixed), behavioral state (sleep, running, and anesthesia), as well as the recording approach and other metadata (e.g., species, sex, age) to ascertain equivalences and differences across conditions. Using oscillatory pieces of knowledge of morphologically identified PV+ interneurons as a first use case, we show how the updated Hippocampome.org can be employed to improve cell-type profiling of optotagged extracellularly recorded cells. We summarize our approach with a discussion of the opportunities and challenges for integrating single-cell phenotypes with microcircuit function.

## Results

### Mapping single-cell oscillatory dynamics to Hippocampome.org cell types

Hippocampome.org v.1.8 [23] defines 122 neuron types from the dentate gyrus (DG), CA3, CA2, CA1, the subiculum, and entorhinal cortex. Every type is characterized by a set of unique properties always including area- and laminar-specific axonal and dendritic patterns as well as its main neurotransmitter. Moreover, Hippocampome.org ascribes neuron types to one of several superfamily categories based on broad commonalities among identifying characteristics. For example, CA1-bistratified cells are GABAergic neurons coexpressing PV, somatostatin (SOM), and neuropeptide-y (NPY); have their dendrites and axons in strata oriens and radiatum, as shown in the Hippocampome.org morphological and neurochemical encodings; are regular-spiking cells; and belong to the Collateral-related cell superfamily (Fig 1A, left; see S1A Fig and S1 Table for general definitions). To complement this knowledge base, we mined the peer-reviewed literature for representative in vivo single-cell firing data meeting the following criteria: (a) cell identity should be validated morphologically; (b) simultaneous local field potential (LFP) recordings should be available; and (c) information about the cell firing pattern during ongoing LFP activity should be reported or inferable from figures and tables.

We retrieved 37 publications meeting these criteria altogether, reporting over 900 relevant data points from 35 entorhinal and hippocampal neuron types (S2 Table). Neurons morphologically identified and/or tested against a battery of neurochemical markers enabled their assignment to specific Hippocampome.org types (Fig 1). Thus, GABAergic interneurons could be ascribed to different subpopulations such as CA1-bistratified cells (Fig 1A; Hippocampome-type CA1 (i)0302 C-Bistratified) and oriens-lacunosum moleculare (O-LM) interneurons (Fig 1B; CA1 (i)1002 O-LM). While most GABAergic interneurons were reported for CA1, we successfully identified matching candidates from other hippocampal regions (Fig 1C; CA3 (i)22232 Basket CCK+). Some glutamatergic cells were annotated to newly defined specific subpopulations, such as superficial CA1 pyramidal cells (Fig 1D; CA1 (e)2223p Superficial Pyramidal), and to medial entorhinal cortex (MEC) layer II stellate cells (Fig 1E; MEC (e)331111p LII Stellate).

We noticed disparate criteria for assigning phases to theta peaks and troughs (e.g., 0° marks the trough in Fig 1C, while in Fig 1D, it marks the peak of the oscillation). This, together with the typical phase reversal of LFP theta cycles across layers and regions, complicated annotation. Thus, we curated all data, cell by cell, to refer phase-locking behavior against a common reference at CA1 stratum pyramidale (SP), by using information on the reported LFP site and theta-peak correction (S1B–S1D Fig; S2 Table, red values). Values reported per cell in tables and/or figures of the original publication were all annotated to the same Hippocampome.org neuron type to enable cross-correlation analysis between indices.

### Evaluation of existing data

In most references consulted, single-cell firing was evaluated against LFP signals recorded at the CA1 SP, more typically during theta oscillations (4 to 12 Hz) and SWR. Neuronal firing during these network events was variable, both within and between cells. While part of this variability may reflect functional effects or different recording conditions (i.e., anesthesia versus drug-free preparations; sleep versus awake), major trends emerged consistent with the idea of cell-type specific firing behavior. For instance, GABAergic CA1-bistratified cells preferentially fire during the ascending ripple phase (Fig 1A; [12]), while O-LM cells tend to decrease their firing rate during SWR as compared with baseline (Fig 1B; [7]); but note heterogeneous participation of these cells during a variatey of SWR [7,13]. Depending on the analytical emphasis and focus of the publication, these firing trends can be summarized differently (e.g.,

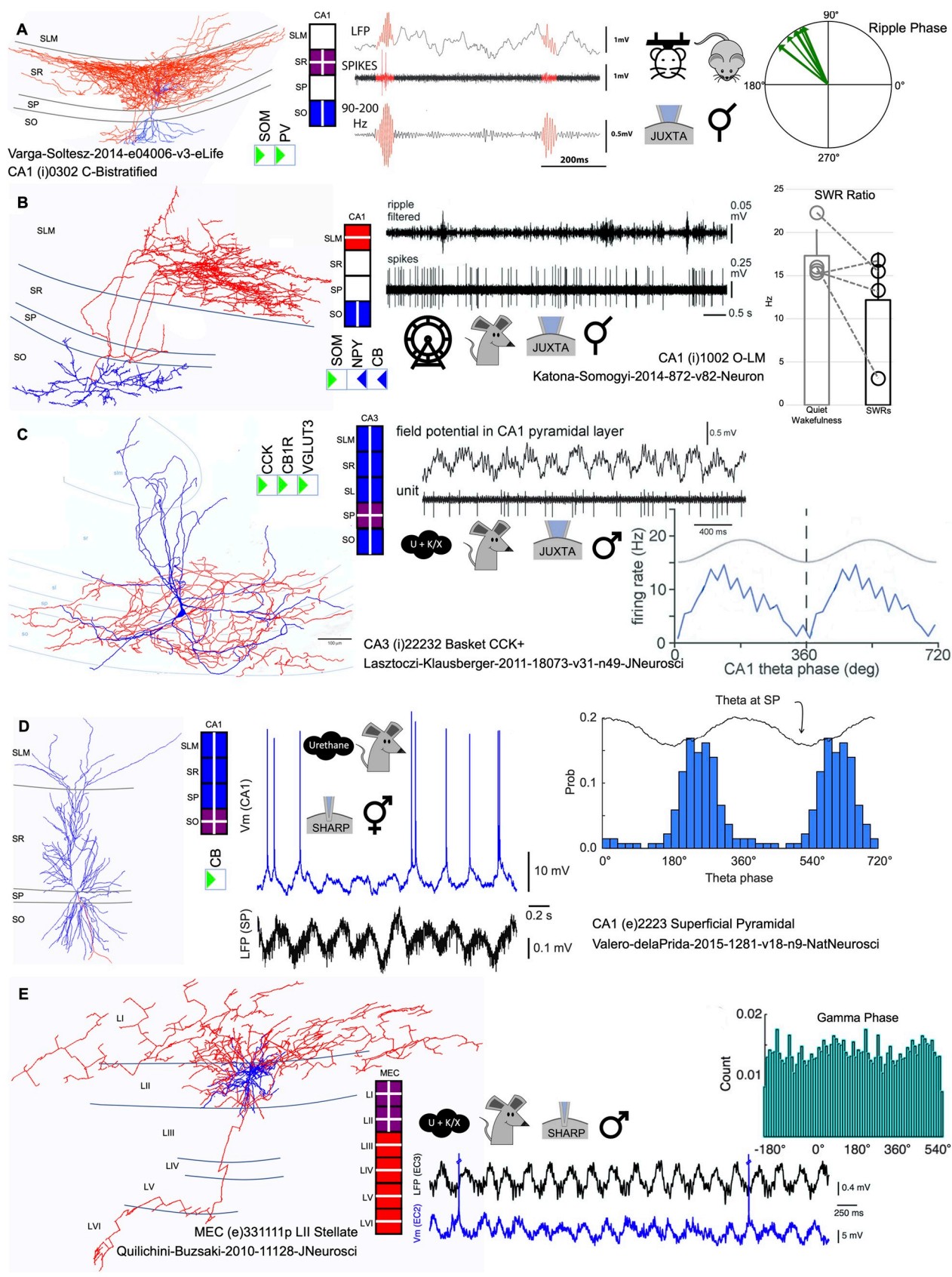

**Fig 1. Diversity of representative in vivo data for identified neuron types of the rodent hippocampal formation in multiple conditions. (A)** A CA1 (i)0302 C-Bistratified cell (soma and dendrites in blue; axon in red) from NeuroMorpho.Org [NMO_61413] [24] with corresponding Hippocampome.org biomolecular markers (green triangle: positive; blue triangle: negative) and morphological encodings (purple: axons and dendrites in layer; blue: dendrites in layer). Traces of LFP, spikes, and ripple oscillations, with 2 high-frequency epochs (highlighted in red) [12] are accompanied by a metadata summary indicating behavioral state (head-fixed awake), animal (mouse), recording method (juxtacellular), and sex (unknown). The polar plot represents ripple phases for CA1 C-Bistratified cells. **(B)** A CA1 (i)1002 O-LM cell from NeuroMorpho.Org [NMO_36625] with Hippocampome.org morphological (red: axon in layer) and biomolecular marker encodings; traces of filtered ripple oscillations and spikes [7]; metadata summary indicating behavioral state (freely moving), animal (rat), recording method (juxtacellular), and sex (unknown); and plot of the SWR ratio, defined as the ratio between the firing rate at the ripple peak over the basal firing rate. **(C)** A CA3 (i)22232 Basket CCK+ cell adapted from [25] with Hippocampome.org biomolecular marker and morphological encodings; metadata summary indicating behavioral state (urethane with supplemental doses of ketamine and xylazine), animal, recording method, and sex (male); firing pattern associated with theta oscillations: field potential (top) and unit firing (bottom); and plot of mean firing rate as a function of theta phase. **(D)** A CA1 (e)2223p Superficial Pyramidal cell from NeuroMorpho.Org [NMO_60491] with morphological and biomolecular marker encodings; intracellular membrane voltage plotted above the associated LFP signal [26]; summary of behavioral state, animal, recording method (sharp intracellular), and sex (male and female); and plot of firing probability vs. theta phase. **(E)** A MEC (e)331111p LII Stellate cell from NeuroMorpho.Org [NMO_07249] with morphological encoding and metadata indicating behavioral state, animal, recording method, and sex. Short epoch (5 s) of LFP in EC layer III (1 Hz to 1.25 kHz) and membrane potential of an EC LII Stellate cell [27] are shown with population discharge probability for gamma phase. CB, calbindin; CCK, cholecystokinin; EC, entorhinal cortex; LFP, local field potential; MEC, medial entorhinal cortex; NPY, neuropeptide-y; SL, stratum lucidum; SLM, stratum lacunosum moleculare; SO, stratum oriens; SP, stratum pyramidale; SR, stratum radiatum; SWR, sharp-wave ripple.

ripple phase versus SWR ratio; Fig 1A and 1B). In contrast, information on firing dynamics during theta oscillations is more consistently reported in terms of phase-locking preference and modulation (Fig 1C), which are also cell-type specific. For example, while CCK+ basket cells from CA3 tend to fire preferentially during the rising phase of theta oscillations recorded at the CA1 SP (Fig 1C), most CA1 pyramidal cells fire near the theta trough (Fig 1D; [6,28]). Finally, single-cell firing dynamics during gamma (25 to 90 Hz) and/or so-called epsilon (90 to 130 Hz) oscillations were reported in a minority of cases (Fig 1E; [27,29]).

Next, we analyzed the relative abundance of available pieces of evidence as a function of behavioral state and recording method. Pieces of evidence refer to multiple citations from which values for each piece of knowledge (e.g., preferred theta phase) can be inferred, so that one or more pieces of evidence exist to support a given piece of knowledge. Similarly, there is one piece of knowledge per neuron type and, therefore, for a given variable (e.g., SWR ratio), we expect as many pieces of knowledge as neuron types being categorized.

Juxtacellular recordings contributed a predominance of data in a diversity of behavioral conditions (anesthesia, head-fixed, and freely moving), followed by sharp intracellular approaches, which were more prevalent under anesthesia (Fig 2A). Single-cell data from mice were underrepresented compared to rats and more typically contributed pieces of evidence in the head-fixed preparation than in freely moving conditions. A large majority of cells were isolated from adult animals, mostly males (Fig 2B). Overall, no single acquisition protocol accounted for more than one quarter of the reported data: The largest contribution came from juxtacellular recordings under anesthesia, which, even together with similar data from the drug-free head-fixed setup, was still outnumbered more than 2-fold by the long tail of all other preparations (Fig 2C). In terms of the types of single-cell phase information gathered, there was a clear predominance of pieces of knowledge about theta and SWR (Fig 2D), including information on preferred theta phase and SWR ratio. When considering hippocampal areas, we observed a net preponderance for CA1, followed by the entorhinal cortex and the CA3 region (Fig 2E), while cells from DG, CA2, and the subiculum (Sub) were underrepresented.

## Complementary in vivo data of underrepresented cell types and missing pieces of knowledge

We obtained additional experimental data in order to supplement information from some relevant neuron types. We noticed that existing Hippocampome.org information on membrane

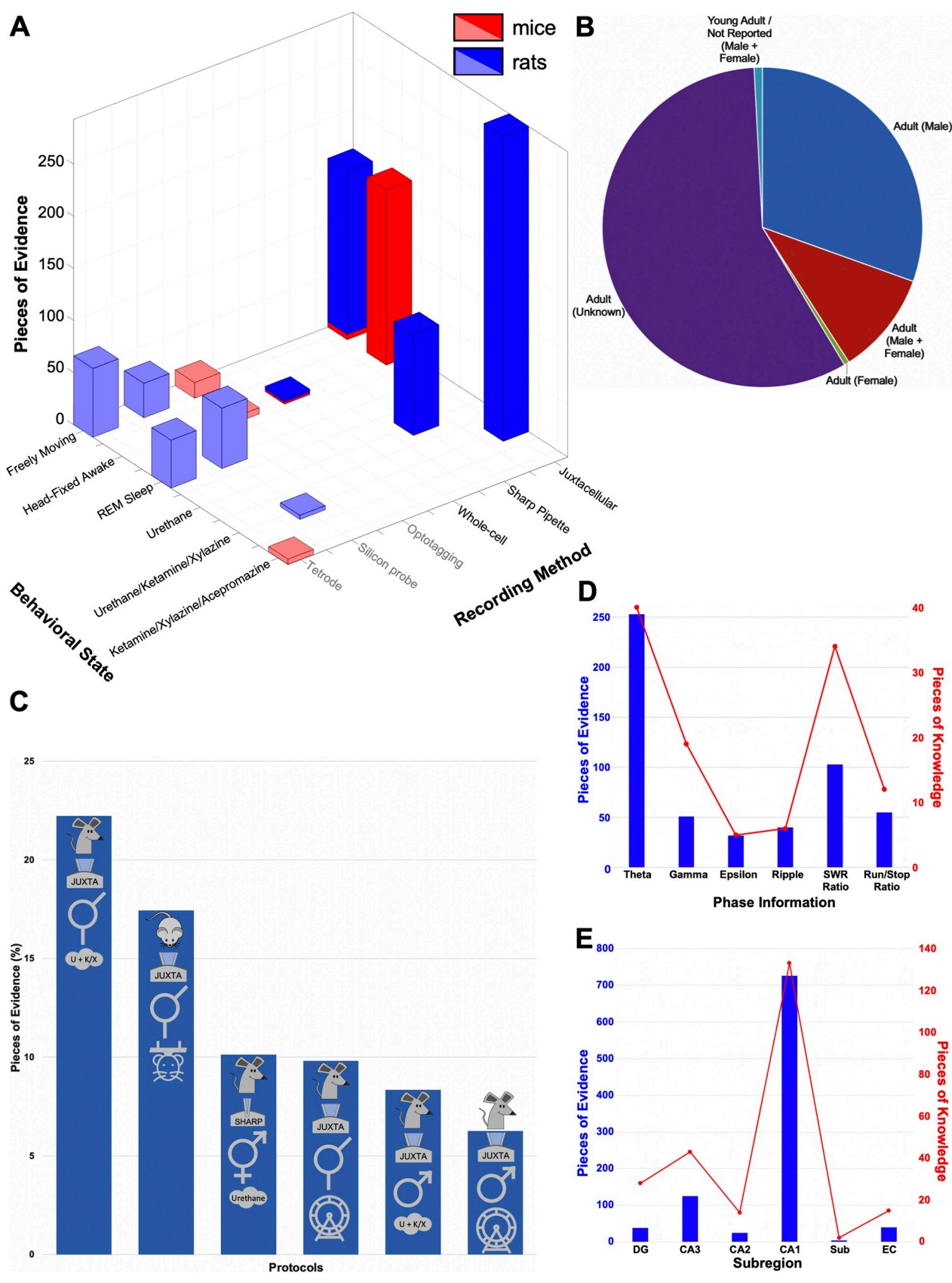

**Fig 2. Summary of existing data and conditions. (A)** Distribution of pieces of evidence relative to behavioral state and recording method (mice in red; rats in blue). Bold colors and black font indicate single-cell recording, and pale colors and gray font indicate extracellular recording. **(B)** Breakdown of pieces of evidence as a function of the age and sex of animals independent on the species. **(C)** Proportion of pieces of evidence by recording protocol for the top 6 protocols. **(D, E)** Numbers of pieces of evidence (in blue) and pieces of knowledge (in red) relative to the types of phase information (D) and the hippocampal subregions (E). The underlying data can be found in S1 Data. DG, dentate gyrus; EC, entorhinal cortex; Sub, subiculum; SWR, sharp-wave ripple.

biophysics and firing patterns were all obtained in vitro from slice preparations. Given their relevance to inform computational models [30], we favor sharp intracellular recordings in vivo to complement these in vitro data. Thus, we obtained new data from cells in the DG ($n = 3$ granule cells and a mossy cell) and CA3 region ($n = 7$), as well as from GABAergic cells ($n = 2$ PV basket cells). In addition, we complemented these new data by fully reanalyzing our own published dataset from CA2 and CA1 pyramidal cells [26,31]. Inclusion criteria were: (a) unequivocal morphological identification; (b) evaluation of basic membrane and intrinsic firing properties; and (c) availability of simultaneous extracellular LFP signals along the CA1 up to DG. We supplied additional evidence of morphologically validated CA2 ($n = 5$) and CA1 ($n = 19$) pyramidal cells meeting these criteria.

We extracted multiple Hippocampome.org pieces of knowledge from these cells. For example, somatodendritic reconstruction and/or immunohistochemical staining against the granule cell marker Prox1 confirmed the identity of the 3 granule cells (2 cells from the upper blade, one from the lower blade) (Fig 3A, top). For the mossy cell, we validated PCP4 as an alternative marker to AMPA receptor GluA2/3 (Fig 3A, bottom), using double immunostaining (Fig 3E). Similar to granule cells [32], the vast majority of GluA2/3 immunoreactive non-GABAergic hilar cells (i.e., mossy cells) were positive for PCP4, both at the upper and lower blades (Fig 3F), with some rostrocaudal differences (Fig 3G). Consistently, we updated the Hippocampome.org neurochemical encodings to reflect this novel information about granule and mossy cells (Fig 3H; boxed markers).

Similarly, we evaluated pyramidal cells from CA3, CA2, and CA1. For CA3, we identified some pyramidal cells at the CA3c region (S2A Fig) while others distributed along the intermediate (CA3b) and distal (CA3a) subregions. For CA2, we noted some inconsistency with calbindin expression annotated in the v1.8 of Hippocampome.org, which was also updated (S2C Fig). For CA1 pyramidal cells, we clarified unsolved issues regarding expression of some markers by introducing 2 new Hippocampome.org neuron types, i.e., Superficial (positive for calbindin) and Deep (negative for calbindin) pyramidal cells (S3A and S3B Fig; unique ID 4098 and 4099, respectively) [26]. The morphological and marker encodings of Hippocampome.org were updated to reflect this new knowledge. The location of the different cell types along the proximodistal axis (i.e., from the DG towards the Sub) was also noted (see S2 Table), given their functional relevance [31,33], for future incorporation into the knowledge base.

We also evaluated the in vivo intrinsic and membrane properties. As for the example of DG cells, the 2 cell types responded to somatic current injection with a transient slow-wave bursting pattern (coded as TSWB in Hippocampome.org; Fig 3B), which matched one of the in vitro firing pattern phenotypes for DG Granule cells but suggested a novel phenotype for DG Mossy cells. This intrinsic bursting capability was consistently reflected in the firing autocorrelogram (Fig 3C), in agreement with data from juxtacellular and optotagged recordings [17,34,35]. Although evaluating intrinsic biophysical properties in vivo may be inaccurate due to interference with ongoing synaptic activity, we estimated values of the resting membrane potential and membrane time constant, as well as the action potential threshold and afterhyperpolarization (AHP), to allow for comparison with Hippocampome.org in vitro data (Fig 3D). Similar data

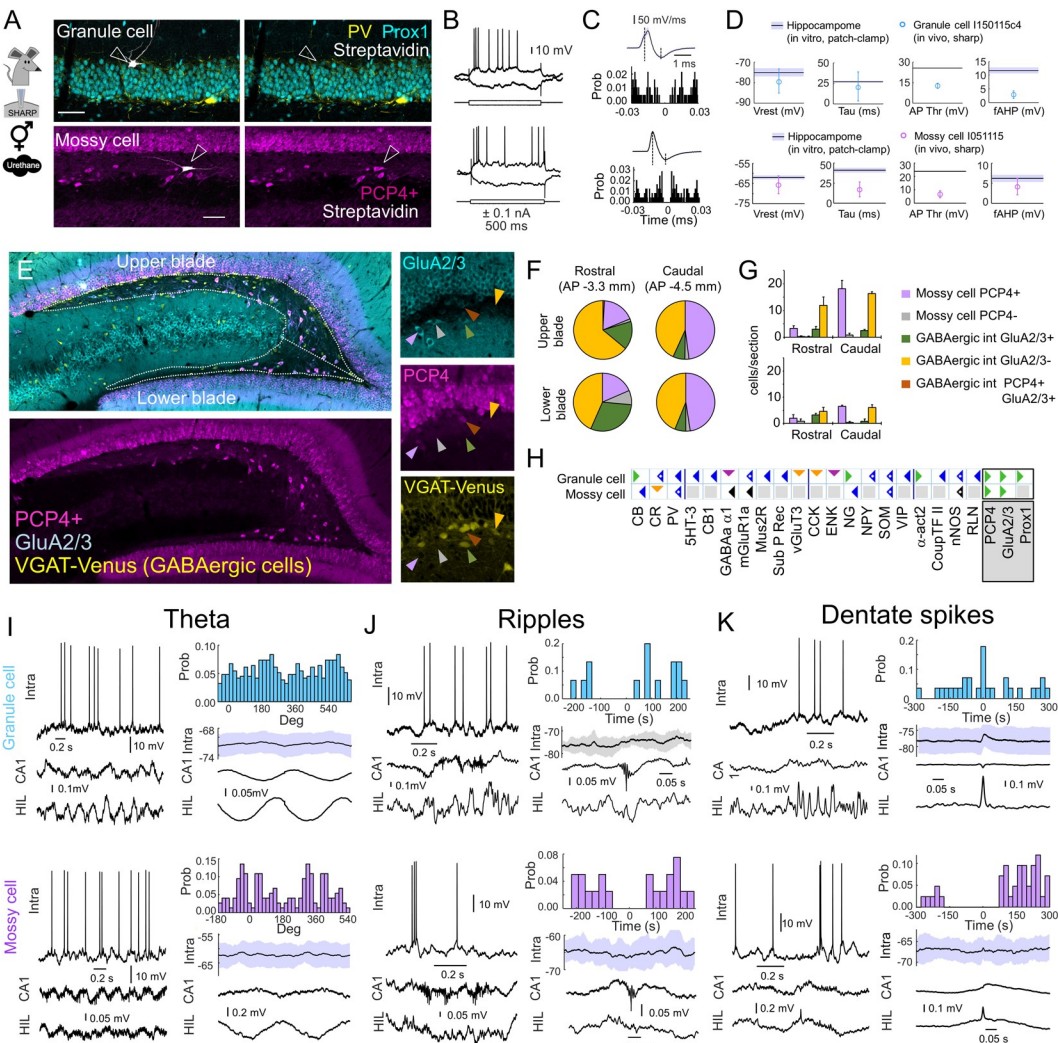

**Fig 3. Oscillatory behavior and neurochemical profiling of DG cells. (A)** Example of a granule cell (lower blade) and a mossy cell (upper blade) recorded intracellularly with sharp electrodes under urethane anesthesia, labeled with Neurobiotin, and identified with streptavidin for immunohistochemical colocalization with Prox1 and PCP4. **(B)** Firing pattern responses of the granule and mossy cells were consistent with TSWB. **(C)** Derivative of the action potential and the firing autocorrelogram of the granule and mossy cells. **(D)** Membrane biophysics properties of granule and mossy cells recorded in vivo as compared with in vitro data from Hippocampome.org. **(E)** Immunohistochemistry against GluA2/3 and PCP4 in VGAT-Venus transgenic rats. The right insets identify the different cell types (arrowheads, see color code in panel G). **(F)** Proportion of cell types as evaluated by complementary expression of PCP4, GluA2/3, and VGAT. **(G)** Quantification of different cell types. Note the colocalization between GluA2/3, PCP4, and VGAT validates bona fide markers of mossy cells. **(H)** Updated Hippocampome.org molecular markers for granule and mossy cells (green triangle: positive; blue triangle: negative; orange triangle: positive–negative due to species/protocol differences; magenta triangle: positive–negative due to subcellular expression differences). **(I)** Firing patterns of the granule cell and the mossy cell shown in A during theta oscillations. Left, Representative traces from each cell. Right, Mean firing histogram (top), membrane potential (middle), and LFP events (bottom) for each cell. **(J)** Same for SWR. **(K)** Same for DS. The underlying data can be found in S1 Data. α-act2, alpha actinin 2; CB, calbindin; CB1, cannabinoid receptor type 1; CCK, cholecystokinin; CoupTF II, chicken ovalbumin upstream promoter transcription factor II; CR, calretinin; DG, dentate gyrus; DS, dentate spikes; ENK, enkephalin; GABAa α1, GABA-a alpha 1 subunit; GluA2/3, AMPA receptor 2/3; HIL, hilus; LFP, local field potential; mGluR1a, metabotropic glutamate receptor 1 alpha; Mus2R, muscarinic type 2 receptor; NG, neurogranin; nNOS, neuronal nitric oxide synthase; NPY, neuropeptide-y; PCP4, Purkinje cell protein 4; Prox1, prospero homeobox protein 1; PV, parvalbumin; RLN, reelin; SOM, somatostatin; Sub P Rec, substance P receptor; SWR, sharp-wave ripple; TSWB, transient slow wave bursting; VGluT3, vesicular glutamate transporter 3; VIP, vasoactive intestinal peptide; 5HT-3, serotonin receptor 3.

**Table 1. In vivo membrane properties for hippocampal cell types (sharp intracellular recordings; values are means ± standard deviations).**

| Hippocampome.org ID | Cell type | Vrest (mV) | Tau (ms) | AP Thr (mV) | fAHP (mV) | AP peak-trough (ms) | n |
|---|---|---|---|---|---|---|---|
| 1000 | Granule cells | −58.5 ± 18.0 | 20.9 ± 13.3 | 7.3 ± 5.1 | 1.15 ± 3.52 | 1.10 ± 0.21 | 3 |
| 1002 | Mossy cells | −65.6 ± 0.0 | 17.3 ± 0.0 | 6.8 ± 0.0 | 3.16 ± 0.00 | 0.76 ± 0.00 | 1 |
| 2000 | CA3 pyramidal | −55.0 ± 6.1 | 28.6 ± 12.9 | 5.2 ± 2.0 | 2.13 ± 1.90 | 0.90 ± 0.12 | 7 |
| 2004 | CA3c pyramidal | −55.0 ± 6.3 | 21.2 ± 5.5 | 5.0 ± 1.9 | 0.96 ± 1.08 | 0.91 ± 0.12 | 4 |
| 3000 | CA2 pyramidal | −51.2 ± 3.2 | 20.3 ± 18.4 | 5.4 ± 2.3 | 2.05 ± 2.03 | 0.87 ± 0.05 | 5 |
| 4000 | CA1 pyramidal | −61.6 ± 5.6 | 18.3 ± 11.3 | 8.8 ± 7.2 | 2.46 ± 2.96 | 0.90 ± 0.11 | 19 |
| 4099 | CA1 pyramidal deep | −58.8 ± 4.4 | 23.9 ± 13.6 | 8.8 ± 7.2 | 4.02 ± 2.98 | 0.91 ± 0.11 | 8 |
| 4098 | CA1 pyramidal sup | −63.6 ± 5.7 | 14.2 ± 7.5 | 10.6 ± 6.0 | 1.33 ± 2.50 | 0.90 ± 0.12 | 11 |
| 4078 | CA1 PV basket cell | −49.1 ± 1.4 | 6.7 ± 1.1 | 3.2 ± 1.8 | 3.97 ± 1.09 | 0.37 ± 0.11 | 2 |

sup, superficial; Vrest, resting membrane potential; AP Thr, action potential threshold; fAHP, fast afterhyperpolarization

from all other supertypes were incorporated into Hippocampome.org (Table 1; S2E Fig for CA2 and CA3 and S3D and S3F Fig for CA1; S4 Fig for CA1 PV basket cells).

Finally, to evaluate single-cell firing dynamics during network events, we detected theta cycles and SWR using information from simultaneous silicon-probe recordings. During theta oscillations recorded at CA1 SP, the firing rate of all cell types was modulated to a different degree, as evaluated with the mean vector length (MVL) (Table 2; Fig 3I for granule and mossy cells; S2F and S2G Fig for CA2 and CA3 cells; S3G and S3H Fig for CA1 pyramidal cells; S4 Fig for CA1 PV basket cells). Interestingly, membrane potential oscillations reliably reflected the rhythmic underlying depolarization (Fig 3I). During SWR, we also found striking differences among cell types. For instance, granule cells tended to be depolarized from the resting membrane potential and fired, whereas the mossy cell was hyperpolarized and silenced (Fig 3J). To facilitate integration with existing literature, we estimated the SWR ratio from the firing dynamics (SWR ratio >1 indicates increase, <1 decrease of activity) and confirmed different trends (Table 2; S2H Fig; S3I Fig). We also evaluated single-cell dynamics during dentate spikes (DS), a prominent hippocampal network event detected at the DG and typically neglected in single-cell firing studies [36]. As expected, granule cells were mostly depolarized and fired, while the mossy cell was typically silenced (Fig 3K), consistent with the characteristic

**Table 2. Network firing properties per hippocampal cell types (sharp intracellular recordings).**

| Hippocampome.org ID | Cell type | Theta phase (°) | MVL | SWR ratio | DS ratio | n |
|---|---|---|---|---|---|---|
| | | | Urethane anesthesia | | | |
| 1000 | Granule cells | 116 ± 39 | 0.25 ± 0.10 | 2.0 ± 1.8 | 11.7 ± 0 (1) | 3 |
| 1002 | Mossy cells | 339 ± 0 | 0.33 ± 0.00 | 0 ± 0 | 0 ± 0 | 1 |
| 2000 | CA3 pyramidal | 181 ± 63 | 0.31 ± 0.13 | 1.1 ± 1.0 | 0.7 ± 0.5 | 7 |
| 2004 | CA3c pyramidal | 168 ± 31 | 0.28 ± 0.12 | 1.7 ± 0.9 | 0.7 ± 0.6 (3) | 4 |
| 3000 | CA2 pyramidal | 184 ± 79 | 0.34 ± 0.13 | 0.5 ± 0.9 | 2.2 ± 1.4 | 5 |
| 4000 | CA1 pyramidal | 206 ± 30 | 0.39 ± 0.14 | 2.7 ± 3.3 | 0.9 ± 1.1 (17) | 19 |
| 4099 | CA1 pyramidal deep | 222 ± 22 | 0.40 ± 0.09 | 0.6 ± 0.5 | 0.50 ± 0.4 | 8 |
| 4098 | CA1 pyramidal sup | 195 ± 32 | 0.39 ± 0.17 | 4.2 ± 3.6 | 1.2 ± 1.4 | 11 |
| 4078 | CA1 PV basket cell | 89 ± 22 | 0.19 ± 0.04 | 4.3 ± 2.1 | 0.4 ± 0 (1) | 2 |
| | | | Awake | | | |
| 4000 | CA1 pyramidal | 45 ± 68 | 0.23 ± 0.05 | 2.5 ± 2.4 | - | 6 |

DS, dentate spikes; MVL, mean vector length; PV, parvalbumin; sup, superficial; SWR, sharp-wave ripple.

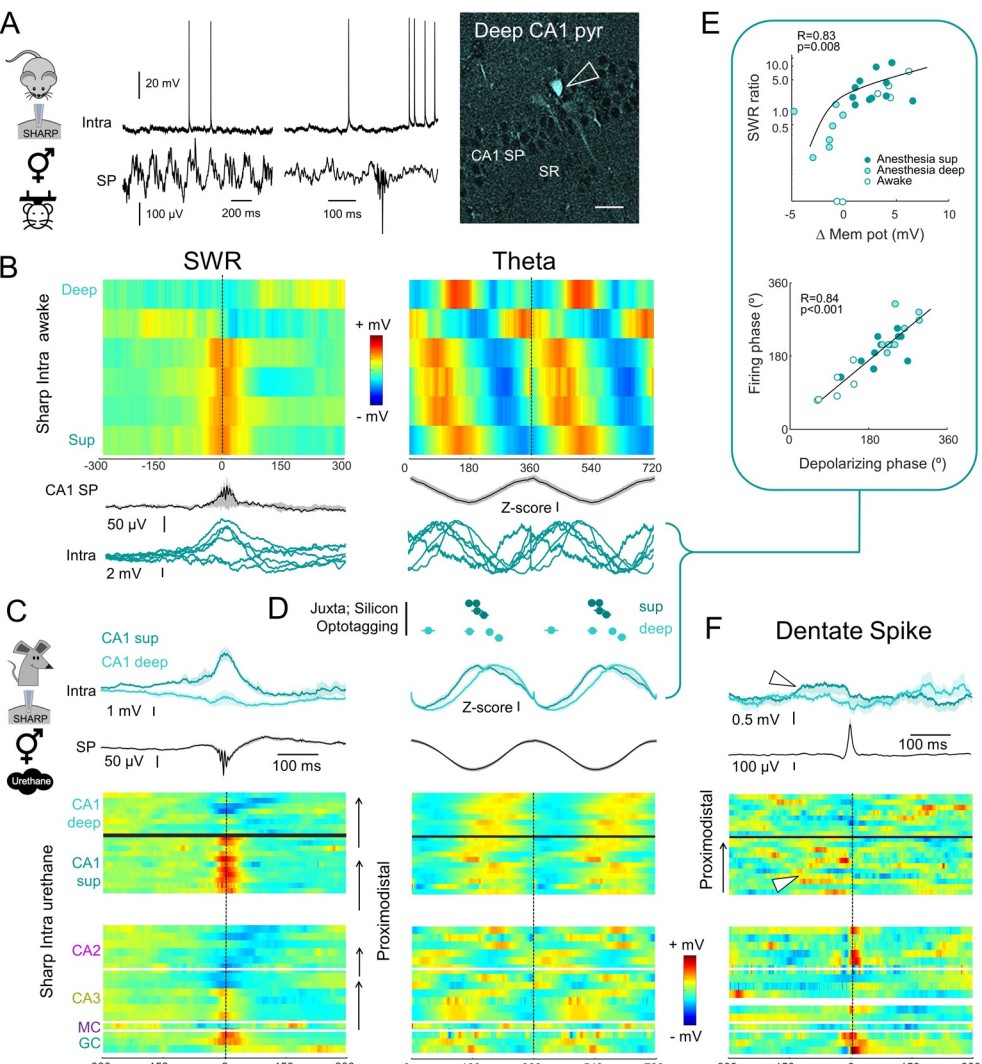

**Fig 4. Comparison across recording conditions. (A)** Intracellular and LFP recordings obtained from awake head-fixed mice. The example CA1 cell was classified as a deep pyramidal cell (right). **(B)** Heatmaps of mean membrane potential responses from all CA1 pyramidal cells recorded in awake head-fixed mice ($n$ = 6), aligned by the SWR peak (left). The morphologically validated deep and superficial cells are plotted at both extremes. Remaining cells were ranked by the mean depolarizing response during SWR. Similar heatmaps for membrane potential fluctuations during theta are shown at right, with cells order as ranked by SWR. The mean LFP event is shown below in black, and SD is depicted in gray. Individual traces from cells are shown in blue. **(C)** Same as before for CA1 pyramidal cells recorded intracellularly with sharp pipettes under anesthesia ($n$ = 11 superficial CA1 pyramidal cells, $n$ = 8 deep CA1 pyramidal cells). Mean ± SEM of membrane potential responses from deep and superficial CA1 cells are shown at top for comparison with awake intracellular data. The rest of cells are shown below (urethane; $n$ = 3 granule cells, $n$ = 1 mossy cell; $n$ = 7 CA3 pyramidal cells; $n$ = 5 CA2 pyramidal cells), all ranked according to their proximodistal location within each region. Note consistent membrane potential gradients across CA3 and CA2. **(D)** Theta phase-locked firing data of deep and superficial CA1 pyramidal cells obtained with juxtacellular electrode [7,28], silicon probe [38,39], and optotagging recordings [28] in awake mice and rats. Note consistent phase-locking across conditions and preparations for superficial cells but not for deep cells. **(E)** Correlation between membrane potential responses and firing rate indices of CA1 pyramidal cells during SWR (top; exponential fitting) and theta (bottom), estimated with sharp intracellular recordings in anesthesia and awake conditions. **(F)** Same as in panels B and C for DS recorded under urethane anesthesia in rats. Cells follow the same proximodistal order as in panel C. Note sharp depolarizing responses at the DS peak in granule cells and CA2 pyramidal cells, and the sustained plateau depolarization in superficial CA1 pyramidal cells (arrowhead). Note also hyperpolarizing responses in deep CA1 pyramidal cells. The underlying data can be found in S1 Data. DS, dentate spikes; LFP, local field potential; SD, standard deviation; SP, stratum pyramidale; SR, stratum radiatum; SWR, sharp-wave ripple.

**Table 3. A summary of some novel discoveries and their potential impact.**

| Key Highlight | Knowledge Gap | Novel Discovery | Practical Outcome | Potential Impact |
|---|---|---|---|---|
| Functional heterogeneity of hippocampal principal cells linked to converging in vivo electrophysiology and molecular expression | The unresolved heterogeneity of CA1, CA3, and CA2 Pyramidal cells hampers understanding of circuitry and data interpretation | Disambiguation of Deep and Superficial CA1 Pyramidal cells, as well as proximodistal CA3 and CA2 neurons | Two new subtypes of CA1 Pyramidal cells, plus CA3c and CA2 cells annotated in Hippocampome.org based on their distinguishing properties | Clarification of the nuanced roles of distinct neuron types in the hippocampal network during awake behavior |
| Comprehensive single-cell phenotyping in vivo | Inconsistent reporting of neuronal firing dynamics across physiological events | Systematically characterized neuron type activity during dentate spikes and through sharp wave ratio | In vivo membrane biophysics, firing frequencies, and a wide range of oscillatory dynamics now available online | New knowledge regarding hippocampal circuit function bridging cellular and behavioral levels |
| Integration of theta oscillation data across anatomical areas and experimental preparations | Theta rhythms often recorded in highly nonuniform conditions and reported in nonstandardized way | Normalized calibration of theta phases throughout hippocampal formation relative to CA1 pyramidal layer | New, behaviorally relevant, reliable traits suitable for identifying neuron types in vivo released in open access | Meaningful interpretation of results across experiments, quantitative between-lab comparisons |
| Novel information on molecular biomarkers for key hippocampal neurons, including gradients | Ambiguous or inconclusive reports of biomarker expression for certain neuron types | New molecular markers identified for DG Granule and Mossy and elucidated expression for CA2 Pyramidal cells; gradients (e.g., nNOS) in CA1 Pyramidal cell subtypes | Richer human- and machine-readable data on neuron type molecular expression added on newly released Hippocampome.org including notations of gradients across known neuron subtypes | Ability to identify neuron types more readily or reproducibly in challenging experimental conditions and accurate recognition of spatial patterns |
| Ground truth for aiding classification of cell types in vivo | Insufficiently specific marker-based neuron classification (e.g., CA1 Parvalbumin interneurons) | Disambiguated extracellular recording of spikes from PV-bistratified, Axo-axonic, and PV-basket cells | Expedient annotation of genetically targeted high-throughput recordings | Enabling high-confidence neuron phenotyping for optotagging and other cell sorting methods |

DG, dentate gyrus; nNOS, neuronal nitric oxide synthase; PV, parvalbumin.

feedforward inhibition in the hilus [37]. Glutamatergic cells along CA3, CA2, and CA1 exhibited variable degrees of firing modulation during DS, as evaluated with a DS ratio, which is defined from the firing rate around the event peak over the basal firing rate (Table 2; S2H Fig; S3I Fig). Individual values from all cell types are included in S3 Table.

## Equivalence of results across recording methods and conditions

A major challenge in interpreting data gathered from the literature is the diversity of recording methods (e.g., sharp intracellular versus juxtacellular electrodes), experimental conditions (anesthesia versus drug-free), and reporting conventions. To help evaluate the equivalence of firing rate data across preparations, we obtained additional sharp intracellular recordings from CA1 pyramidal cells in head-fixed awake mice and simultaneously LFP signals at CA1 SP ($n = 6$ cells; 2 morphologically identified; Fig 4A). We then measured their membrane potential dynamics during SWR and theta oscillations (Fig 4B) and proceeded similarly with CA1 pyramidal cells recorded intracellularly under urethane (Fig 4C).

We noted similar SWR profiles in awake versus anesthetized, with some pyramidal cells mostly depolarized while others exhibited a hyperpolarizing trend at the event peak (Fig 4B, left; U(19, 6) = 60.0, $p = 0.8770$, Mann–Whitney $U$ test). Under urethane, part of this behavior can be explained by the deep/superficial location of the recorded cell within the CA1 layer (Fig 4C; mean membrane potential traces from deep and superficial cells are shown at top; SWR index, U(11,8) = 1, $p < 0.0001$, Mann–Whitney $U$ test; S3J Fig) [26].

In contrast, during theta, there were major differences between conditions (U(19,6) = 1, $p = 0.00002$, Mann–Whitney $U$ test), possibly reflecting the contribution from different oscillators [40]. Under urethane, the membrane potential depolarization from deep and superficial

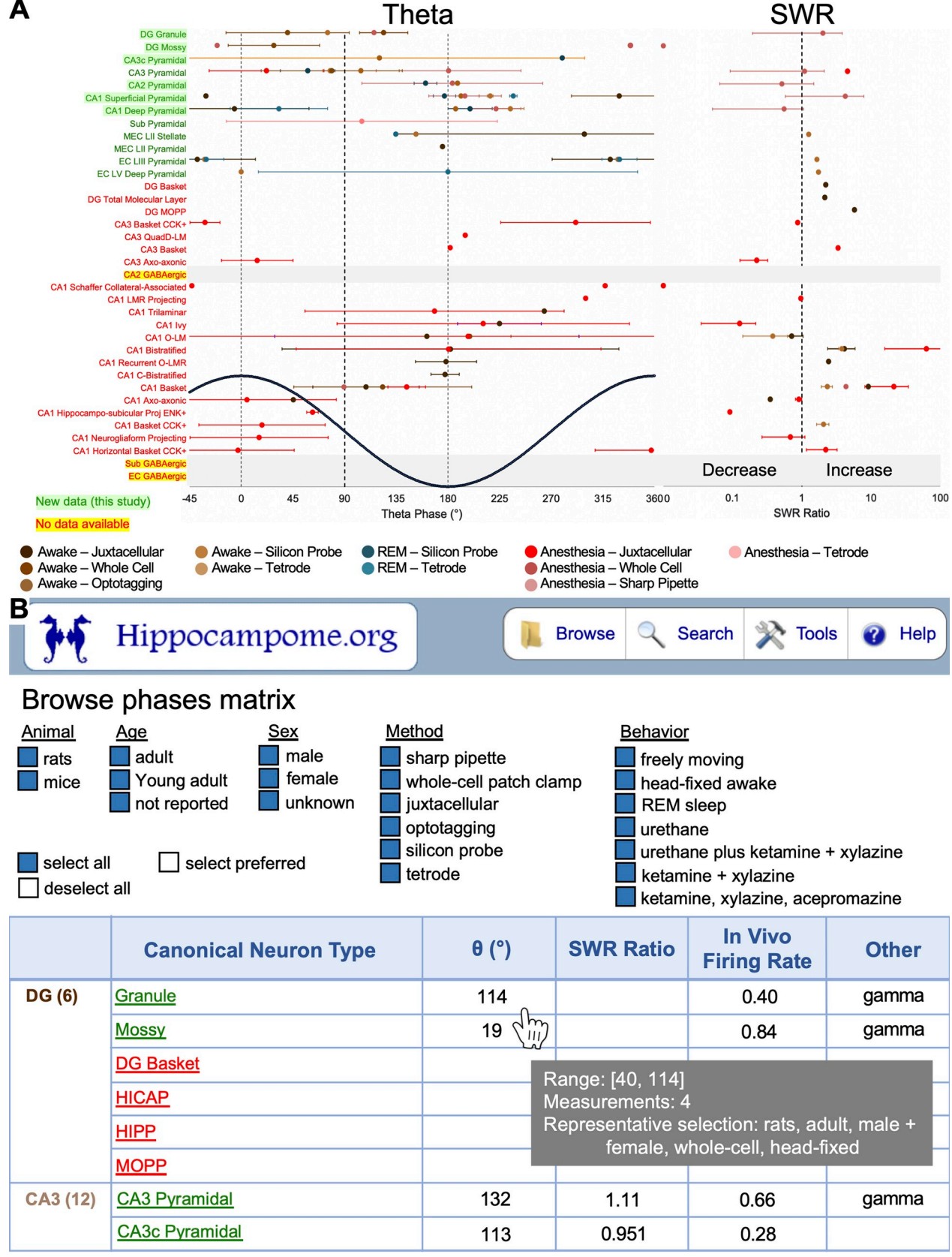

**Fig 5. Summary of Hippocampome.org in vivo firing dynamics data.** (**A**) Summary of literature-mined information comparing in vivo phase-locking behavior of identified Hippocampome.org neuron types (glutamatergic in dark green, GABAergic in red) with respect to theta oscillations and SWR ratio. New data from this study are highlighted in light green. Cell types with no available data in the literature are highlighted in yellow. (**B**) Presentation of theta phases, SWR ratios, and other in vivo firing data on the Hippocampome.org website. Users are able to select combinations of metadata for displaying data values in a summary matrix. Hover-over windows present the user with information pertaining to the range of data values available given the metadata selections, the number of data values available in the literature given the metadata selections, and a description of the preferred metadata combination used to select the value presented in the matrix. Clicking on any of the matrix values navigates to an Evidence page, which contains the literature references supporting the data values. The underlying data can be found in S1 Data. CCK, cholecystokinin; DG, dentate gyrus; EC, entorhinal cortex; ENK, enkephalin; LMR, lacunosum moleculare-radiatum; MEC, medial entorhinal cortex; MOPP, Molecular layer Perforant Path-associated; O-LM; O-LMR, oriens-lacunosum moleculare radiatum, oriens-lacunosum moleculare; QuadD-LM, quadrilayer dendrites-lacunosum moleculare; REM, rapid eye movement; SWR, sharp-wave ripple.

CA1 pyramidal cells tended to peak slightly differently, which affected when they fired ($t(19) = 2.3$, $p = 0.0346$, Student $t$ test; Table 3; Fig 4C, right; S3I Fig). Superficial cells were maximally depolarized at the theta trough ($195 \pm 32°$), while deep cells were shifted towards the ascending theta phase ($222 \pm 22°$). In head-fixed running mice, variability across intracellularly recorded cells was pronounced (Fig 4B, right; see individual traces). Integrating theta phase-locked firing information from juxtacellular and optotagged recordings of morphologically confirmed deep and superficial CA1 pyramidal cells in vivo [28,38] supports the notion that membrane potential variability likely reflected sublayer trends (Fig 4D). This correspondence between neuronal firing and membrane potential dynamics was further supported by significant correlation between the indices estimated from sharp intracellular recordings in both conditions (Fig 4E).

Our data also highlighted major differences of membrane potential dynamics across cell types and events (Fig 4C), in agreement with firing rate information from multisite silicon probe recordings in freely moving rats [33,41]. In addition, we uncovered striking region and cell-type specific biases during DS (Fig 4F), with brief depolarizing responses in granule and CA2 pyramidal cells, as expected from direct layer II entorhinal inputs [42]. More remarkably, we found significant depolarizing plateaus in superficial CA1 pyramidal cells preceding DS peaks at the DG and a transient hyperpolarizing trend in some deep cells (Fig 4F, arrowheads). Altogether, these data reflect the importance of leveraging on distinct cell-type specific in vivo functional features to improve classification of single cells.

## Updating Hippocampome.org to integrate in vivo single-cell firing dynamics

Next, we updated Hippocampome.org cell types to integrate new and existing information on in vivo firing behavior during theta oscillations and SWR (Fig 5A). We included all data gathered from the literature and new data reported/analyzed in this study.

Mean and standard deviation data are reported in Fig 5 for theta and SWR. This representation highlighted striking differences across cell types and regions during both types of network activities. Some neuronal types not included in v1.8 of Hippocampome.org, such as deep and superficial CA1 pyramidal cells, were incorporated as new subtypes (Fig 5A). New knowledge regarding in vivo firing dynamics of underrepresented cells, such as granule cells and mossy cells of the DG, was also included, together with updated molecular markers (GluA2/3 and PCP4). To bring the attention of the neuroscience community to missing pieces of knowledge, we highlighted categories of neuron types with no available data in the existing literature (Fig 5A, yellow). Metadata indicating behavioral state (awake, sleep, anesthesia), recording method (juxtacellular, whole-cell, sharp, cell-type specific optotagging, silicon probe, tetrodes), animal (rat, mouse), and sex were all integrated to facilitate the exploration of different biases (see color code in Fig 5A). Finally, the Hippocampome.org website was updated by including a

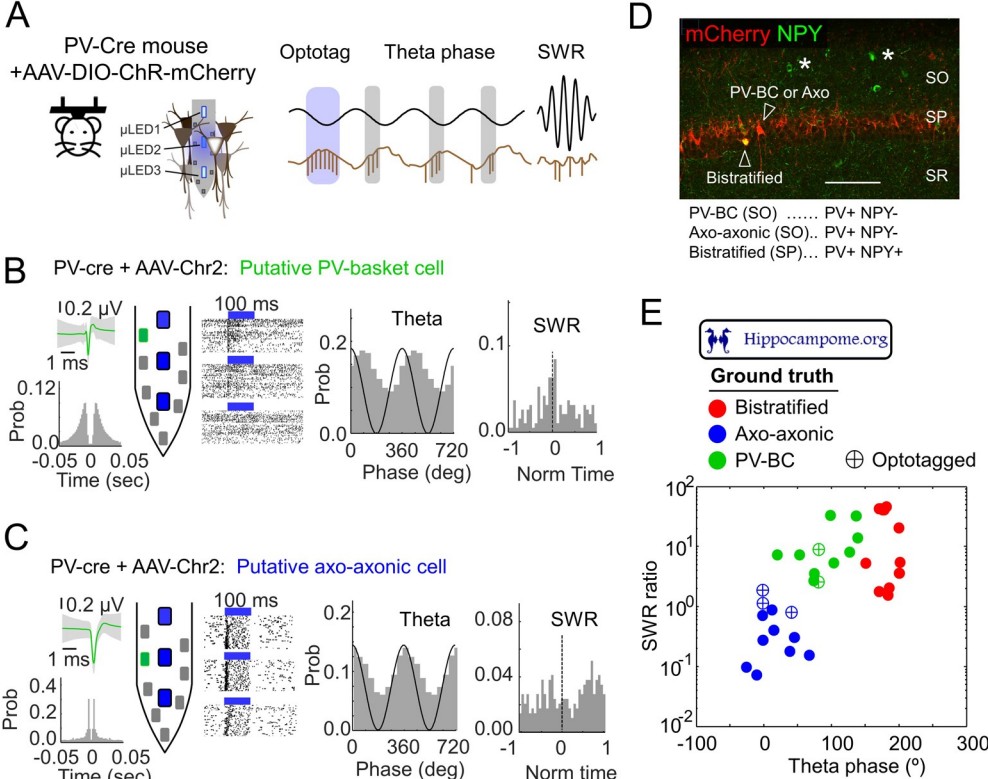

**Fig 6. Using Hippocampome.org to promote knowledge-based phenotyping of optotagged cells in vivo. (A)** PV-cre head-fixed mouse injected with AAV-DIO-ChR2-mCherry to couple optogenetic tagging with high-density recording. (**B**) Example of one optotagged putative PV-BC. The extracellular action potential waveform and autocorrelogram of the unit is shown at left. The channel corresponding to the maximal amplitude action potential is shown green in the probe drawing (i.e., channel closest to the unit). Responses of the unit to several trials of optical stimulation is depicted next to the probe for each of the 3 LEDs, followed by the unit theta phase-locked firing and responses during SWR. (**C**) Example of one putative axo-axonic cell. Same as in B. (**D**) Histological confirmation of the subpopulations of CA1 GABAergic cells targeted in the PV-cre mouse. Different PV+ cell types are indicated by arrowheads. Asterisks indicate some NPY+ cells. (**E**) The firing dynamics of $n$ = 5 optotagged units during theta and ripples are evaluated and confronted against ground truth from the Hippocampome.org phase encoding. The location of optotagged cells with respect to cell-type specific clusters is evaluated with k-means: putative PV-BC were significantly closer to the PV-BC cluster ($p$ = 0.0003); putative axo-axonic cells were significantly closer to the axo-axonic cluster ($p$ = 0.0004). The underlying data can be found in S1 Data. LED, light-emitting diode; NPY, neuropeptide-y; PV, parvalbumin; PV-BC, PV basket cell; SO, stratum oriens; SP, stratum pyramidale; SR, stratum radiatum; SWR, sharp-wave ripple.

new browsable matrix with detailed information of multiple pieces of knowledge related to in vivo firing dynamics (Fig 5B).

To evaluate the added value of the new data, we ran a pairwise correlation analysis with categorical contingency matrices as well as an analysis of Spearman's rank correlation with discrete ordinal and continuous values. We investigated 248 properties, which included morphology, biomolecular markers, in vitro membrane biophysics and firing patterns, single-compartment Izhikevich modeling parameters, and in vivo theta phase locking, SWR ratios, and firing rates. We opted to analyze the data using contingency matrices, since a majority of the pieces of knowledge, such as the presence or absence of axons and dendrites in the 26 layers of the hippocampal formation and the clear positive or negative expression of biomolecular markers, are nominally categorical in nature, as characterized in Hippocampome.org. To facilitate the analysis of all properties of the neuron types collated in Hippocampome.org, those

data that are continuous in nature were converted into categorical properties, such as evaluating the top one-third and bottom one-third of values of the 10 membrane biophysics properties that were examined. We chose Spearman's analysis, as opposed to Pearson's correlation, because not only did we evaluate continuous values, such as the membrane biophysics properties, but we also investigated discrete ordinal values, such as the presence or absence of neurites in a given layer of the hippocampal formation, which Pearson's is unsuitable to quantify. Using both the categorical and Spearman's analyses, we reached the same conclusion that none of the 3 new in vivo properties exhibited a significant correlation with any of the in vitro Hippocampome.org properties (all pairwise comparisons $p > 0.05$). These results suggest that there might not yet be enough in vivo data to fully assess such correlation analyses, but the results also hint at the prospect that these in vivo properties could add classificatory power, when combined with traditional in vitro characterization, in the evaluation of in vivo extracellular recordings.

## Using Hippocampome.org data to promote knowledge-based discoveries and analysis

Finally, we looked to apply the updated Hippocampome.org to assist in the identification of optotagged extracellularly recorded cells, a critical issue for circuit mapping. To illustrate how Hippocampome.org can be further exploited to this purpose, we used head-fixed PV-Cre mice injected with AAV-DIO-ChR2 and recorded with high-density micro-light-emitting diode (LED) optoelectrodes (Fig 6A). Coupling transgenic Cre lines with optogenetics and high-density recordings allows the achievement of specific control of genetically defined neuronal populations [43]. However, this approach selectivity relies on genetic promoters, which in many cases are heterogeneous in a diversity of cell types, and therefore ambiguity persists.

We optotagged 5 units from 2 mice. Two units were identified as putative PV basket cells (PV-BC) based on their preferred firing along the falling theta phase and strong modulation during SWR (Fig 6B). The other 3 units fired at the theta peak and were only mildly modulated by SWR (Fig 6C). Post hoc immunostaining of the sections containing the probe track confirmed, we were targeting a variety of PV+ interneurons, including PV-BC, axo-axonic, and possibly bistratified interneurons (Fig 6D). Anchoring our recordings to Hippocampome.org in vivo firing data, we looked to assign extracellularly optotagged cells to specific GABAergic interneuron subtypes (Fig 6E; 28 labeled PV+ interneurons with available data on theta phase and SWR ratio). When mapped against the knowledge base, the 2 putative PV+ basket cell optotagged units laid nearest to the PV-BC cluster than to any other ($p = 0.0003$), while the remaining 3 units were assigned to the axo-axonic cluster (k-means; $p = 0.0004$; Fig 6E), confirming the utility of the approach in refining classification of optotagged cell types.

## Discussion

The analysis of in vivo firing dynamics accumulated through literature mining and the evaluation of new and reprocessed data led to several new discoveries and considerable enrichment of Hippocampome.org (Table 3). We uncovered the disambiguation of hippocampal principal cells according to their in vivo electrophysiology and biomolecular marker expression. This could facilitate a nuanced understanding of their distinct roles in the hippocampal network during awake behavior. More consistent reporting of hippocampal firing dynamics across physiological events, such as DS, reveals putative circuit function bridging cellular and behavioral levels. We have established a normalized calibration of theta phases relative to the CA1 pyramidal layer, enabling more meaningful interpretations of results across experiments and between labs. To more readily and reproducibly identify neuron types under challenging

experimental conditions, we have reported on new, less ambiguous biomarker expression patterns for some neuron types and taken note of more graded expression of other biomarkers that are indentified between neuronal subtypes. Finally, we utilize the ground truth of in vitro knowledge accumulated in Hippocampome.org as a means of enabling high-confidence neuron phenotyping in extracellular in vivo experiments.

The previous version of Hippocampome.org included 122 hippocampal and entorhinal cortex cell types [22] classified according to the main neurotransmitter, axonal and dendritic patterns, synaptic specificity [44], molecular biomarkers [45,46], membrane biophysics, and firing patterns [47]. Moreover, this knowledge base also associates each identified neuron type with known information on potential connectivity [48] and synaptic signals [49]. While information about action-potential waveforms and neuronal burstiness may assist classification [50–52] and investigations of spike initiation mechanisms [53], the capability to disambiguate between cell types based on these features remained suboptimal [54]. For example, for some of these neurons, there was uncertainty regarding the neurochemical data, which prevented appropriate classification. For CA1 pyramidal cells in particular, the expression of calbindin, CCK, nNOS, CB1 receptors, and several GABAa receptor subunits were unresolved in earlier versions of Hippocampome.org due to disparate reports possibly revealing natural variability, species-specificity, longitudinal expression gradients, subcellular localization, or differences in the detection protocol (RNA versus protein). We now clarify some of these ambiguities by splitting the population into deep and superficial subtypes: Superficial cells are calbindin positive and express more of the neuronal gene *Nos1*, while deep cells are enriched in transcripts for the GABAa alpha 1 receptor subunit (*Gabra1*) [55] (Table 3). For other neuron types, such as DG, CA3c, and CA2 cells, we have added new knowledge and updated Hippocampome.org neurochemical encodings accordingly [26,31] (Table 3). Data on membrane biophysics and basic intrinsic properties evaluated from in vivo recordings were also annotated to facilitate investigations across preparations.

Emerging single-cell RNAseq data suggest heterogeneous gene expression in molecularly defined populations, which, when combined with electrophysiological characterizations, may enable better understanding of neuronal identity [56,57]. The seminal discovery that in vivo single-cell firing dynamics during ongoing oscillations provide additional experimental evidence for cell-type identification [6,7] offered important new perspectives. Linking membrane-potential fluctuations and firing dynamics provides functionally relevant insights into the underlying mechanisms of circuit operation and helps in constraining computational models [26,31,58–60]. Thus, we chose to systematize the functional profiling of morphologically labeled hippocampal/entorhinal cells in vivo, in particular during theta activity and SWR, to gain additional classificatory power. These 2 forms of oscillatory activity not only provide ground truth to supervise the classification of extracellularly optotagged cells but may also foster new hypotheses and facilitate the interpretation of data emerging from ultrahigh density recordings [4,61] (Table 3). Increasing access to multimodal information of morphologically identified cell types is essential to accelerate in vivo imaging and annotation of a diversity of cell types in real time [18] as well as to link the resulting data to a wealth of intrinsic cellular properties derived from slices in vitro.

By integrating information about neuronal identity, location, morphology, and firing dynamics, we increase our capability to understand hippocampal–entorhinal function. For instance, the analysis of membrane potential dynamics of identified cells from the DG to CA1 suggests striking regional organization across the proximodistal and deep-superficial axes both in awake and anesthetized conditions (Table 3). During SWR under urethane, neuronal activity runs as an avalanche from the proximal CA3c mostly through superficial CA1 cells, while the activity of distal CA3a region and CA2 pyramidal neurons is mostly decreased [26,62,63].

At CA3c, interactions with hilar cell types and DG granule cells may be playing roles in shaping SWR initiation [64–66]. Instead, during awake/sleep transitions, a dynamical switch between CA3c and CA3a/CA2 interactions looks critical to shape firing selection [41,67]. Moreoever, the observation of interacting dynamics between different forms of activity such as SWR, DS, and cortical up/down states provided additional insights. Our data suggest that during DS, activity remains regionally constrained at the DG, CA2, and the superficial CA1 sublayer, pointing to the critical role of subcircuit-specific routing of information during different hippocampal network activities [19,68,69] (Table 3).

The categorical and Spearman's analyses did not exhibit any significant linear or monotonic pairwise correlations, respectively, between the in vivo properties examined here and the in vitro properties already existing in Hippocampome.org. This is not to suggest that no relationship could ever be found between these 2 forms of data. In fact, a relationship of this kind is likely to exist, as both forms of data have now been shown to be directly relatable to hippocampal neuron types. However, the available evidence is still too sparse to enable the prediction of the in vivo properties from the in vitro ones.

The compendium of in vivo firing behavior that has been analyzed in this work and added to Hippocampome.org is unprecedented for any mammalian brain region. Continuing to update Hippocampome.org with information on single-cell oscillatory dynamics in vivo should further enable quantitative analyses and data-driven computational simulations, leading to novel discoveries and theories. By illustrating how knowledge-based exploration can lead to augmented understanding of entorhinal–hippocampal function, we hope to encourage the neuroscience community to contribute complementary knowledge to this open-science resource.

## Methods

### Searching criteria and data mining

In the search for pertinent literature, we started with a recent review [19] and backtracked through the citations to the original articles. We also mined early articles for in vivo phase-locking information for CA1 neuron types [6,29,70,71] and more recent publications obtained in drug-free conditions [7,11]. We then proceeded to search for articles that cited these works and performed PubMed searches for each of the subregions of the hippocampal formation and the entorhinal cortex, looking for phrases such as "theta," "gamma," and "ripple." Finally, we updated the search to incorporate any missing references reporting in vivo single-cell firing where: (a) the cell identity was morphologically validated [72]; (b) the cell was recorded simultaneously with respect to LFP signals; and (c) information about the cell intrinsic firing pattern and during ongoing LFP was reported. We data mined articles that quoted phase-locking values, contained tables of values, or depicted phase-locking behavior in diagrams or figures.

We consolidated all data from the different neuron types resulting in 30 hippocampal and 5 entorhinal annotated neuron types (S2 Table). For a minority of neurons, either there was not enough identifying information or the neuron types did not yet qualify for formal inclusion in Hippocampome.org, and they were not included (15 cells). Whenever available, the degree of firing modulation during theta, as well as phase-locking behavior during gamma and epsilon oscillations, were included in the knowledge base. We also include firing rate values and all available indices to characterize single-cell dynamics during SWR and the effect of run/stop transitions from drug-free recordings (S2 Table).

In order to facilitate extrapolation to extracellularly recorded units, we searched for data obtained with multielectrodes. Inclusion criteria were the following: (a) they were obtained with multisite silicon probes, so that at least some morphological information was available

(i.e., laminar/regional location of units); (b) unit identity was validated by cell-type specific optotagging; and/or (c) they were obtained in combination with any of the aforementioned approaches (including glass electrode recordings). Information about recording methods and condition, as well as species, age, and preparation were all included as metadata (S2 Table).

## Data processing and curation

To curate data gathered from the literature, we first established the equivalence between the reported theta phase in a given layer/region and that at CA1 SP (S1B Fig). To this purpose, we considered mean phase shifts estimated across layers (S1C Fig) as follows: DG cell layer −160˚, hilus −170˚, CA3a −30˚, CA3b −80˚, CA3c −180˚ [8], CA2–10˚ [31], CA1 SLM −180˚, and CA1 SR −60˚. For entorhinal cells, phases were corrected only for layer I (−180˚), if available (S1D Fig). In all cases, the CA1 SP theta peak was set at 0˚.

For gamma and epsilon oscillations, as well as SWR, no correction was implemented, as there is no clear equivalence between these local rhythms across layers and regions. In all cases, we gathered data directly from the literature and/or evaluated the data from figures using the freeware program Plot Digitizer (plotdigitizer.sourceforge.net). Where possible, we digitized all available points in polar plots and histograms in order to compute mean phase values and mean vector lengths. These values were calculated similarly as for our own data analysis to facilitate comparison.

## Single-cell recordings in vivo

To complement missing and unresolved pieces of knowledge, we obtained new data from glutamatergic cells from the DG and CA3 and GABAgergic interneurons, using sharp intracellular recordings from male and female adult rats under urethane. We also reanalyzed data from previously published CA2 and CA1 pyramidal cells [26,31]. Only morphologically validated cells, recorded simultaneously to CA1 and DG LFP signals, were considered for analysis.

For sharp intracellular recordings under urethane, adult rats were anesthetized (1.2 g/kg, i. p.), fastened to the stereotaxic frame, and kept warmed (37˚ body temperature). Bilateral craniotomies were performed for stimulation and recordings (AP: −3.7 mm; ML: 3 mm). The dura was gently removed and covered with agar 3%, and the cisterna magna opened and drained. We used sharp pipettes (1.5 mm/0.86 mm outer/inner diameter borosilicate glass; A-M Systems, Carlsborg, WA, USA) filled with 1.5 M potassium acetate and 2% Neurobiotin (Vector Labs, Burlingame, CA, USA). Intracellular signals were acquired with a dedicated amplifier (Axoclamp 900A, 100× gain). The resting potential, input resistance, and amplitude of action potentials were monitored all over experiments. Simultaneous LFP recordings were obtained with 16-channels linear arrays (Neuronexus; 100 μm resolution; 413 μm$^2$ electrode area). LFP signals were preamplified (4× gain) and recorded with a multichannel AC amplifier (100×, Multichannel Systems, Reutlingen, Germany) with analog filters (1 Hz to 5 kHz). Intracellular and LFP signals were sampled at 20 kHz/channel with 12 bits precision (Digidata 1440; Molecular Devices, San Jose, CA, USA). After experiment, Neurobiotin was ejected using 500 ms depolarizing pulses at 0.5 to 2 nA at 1 Hz for 10 to 45 min, and the offset confirmed once the cell was unimpaled. Rats were perfused with 4% paraformaldehyde and the brain cut in 50 to 70 μm coronal sections for posterior histological studies.

To compare against sharp intracellular recordings of CA1 pyramidal cells under anesthesia, we obtained data from drug-free head-fixed mice running freely on a rotary platform. To this purpose, animals were first implanted with fixation bars under isoflurane anesthesia (1.5% to 2%) in oxygen (30%). After surgery, mice were habituated to the apparatus and trained to feel comfortable for up to 2 h with periods of running and immobility. Mice were then

anesthetized to open a craniotomy for electrophysiological recordings. A subcutaneous Ag/AgCl wire was implanted in the neck as reference. One day after this surgery, animals were fixed to the apparatus and intracellular signals were obtained with sharp pipettes as above. Intracellular signals were acquired simultaneously to LFP with multisite silicon probes (Neuro-Nexus, Ann Arbor, MI, USA, and Cambridge Neurotech, Cambridge, England). LFP signals were recorded with a 32-channel AC amplifier (100×, Intan Technologies, Los Angeles, CA, USA) with analog filters (1 Hz to 5 kHz). Single cells were sampled at 20 kHz/channel with 12 bits precision (Power1401-3A; CED). After experiment, Neurobiotin was ejected similarly as explained before and the animal perfused for posterior anatomical analysis.

To illustrate how updates to Hippocampome.org can assist in the classification of extracellularly recorded cells, we obtained data from $n = 2$ PV-cre mice (B6;129P2-Pvalbtm1(cre) Arbr/J, Jackson Stock No: 008069). The dorsal CA1 region was injected with 1 µl of AAV5--DIO-EF1a-hChR2-mCherry (titer 5.2 $10^{12}$ vg/ml; provided by UNC Vector Core; Deisseroth stock) to constrain expression of ChR2 and a reporter in PV GABAergic interneurons. After 3 weeks, the animal was implanted with fixation bars, as above, and recorded head-fixed with integrated micro-LED optoelectrodes (NeuroLight Technologies, LLC, Ann Arbor, MI, USA). Signals were recorded with AC multichannel amplifiers (RHD2000 Intan USB Board running under Open ephys). Characteristic features such as the laminar profile of theta and SWR, as well as unit activity were used to guide penetration. Optical stimulation were delivered with the OSC1-36 driver system (NeuroLight Technologies) to trigger the micro-lEDs independently. In all cases, the probe position was histologically confirmed.

All protocols and procedures were performed according to the Spanish legislation (R.D. 1201/2005 and L.32/2007) and the European Communities Council Directives (2003/65/CE) and were approved by the Ethics Committee of the Instituto Cajal (CSIC) (approval # PROEX 131/16). Experiments leading to intracellular data from awake head-fixed mice were performed in the NYU Institute of Neuroscience and approved by the Institutional Animal Care and Use Committee (IACUC) at New York University Medical Center (approval #160926).

### Histology and immunostaining

To validate recorded and labeled cells, animals were perfused with 4% paraformaldehyde and 15% saturated picric acid in 0.1 M (pH 7.4) phosphate buffered saline (PBS). Brains were post-fixed and serially cut in 50 to 70 µm coronal sections (Leica VT 1000S vibratome). Sections containing Neurobiotin-labeled cells were localized by incubation in 1:400 Alexa Fluor488--conjugated streptavidin (Jackson ImmunoResearch 016-540-084) with 1% Triton X-100 in PBS (PBS-Tx) for 2 h at room temperature (RT).

Sections containing the somata of recorded cells were treated with Triton 1% and 10% fetal bovine serum (FBS) in PBS. Sections were incubated overnight with the primary antibody solution containing any of the following antibodies in 1% FBS in PBS-Tx: rabbit anti-calbindin (1:1,000, CB D-28k, Swant CB-38); mouse anti-calbindin (1:1,000, CB D-28k, Swant 300); mouse anti-Prox1 (1:500, Sigma MAB5654); rabbit anti-Somatostatin-14 (1:2,000, Peninsula T-4103), rabbit anti-Neuropeptide Y (1:1,000, Peninsula T-4070), mouse anti-PV (1:4,000, Swant PV-235), goat anti-mcherry (1:200, Sicgen AB0040), rabbit anti-PCP4 (1:100, Sigma HPA005792), mouse anti-PCP4 (1:100, Novus Biologicals NBP2-61410), and rabbit anti-GluA2/3 (1:300, Upstate 07–598). After 3 washes in PBS-Tx, sections were incubated for 2 h at RT with secondary antibodies: donkey anti-rabbit Alexa Fluor488 (1:200, Invitrogen, A-21206), goat anti-mouse Alexa Fluor488 (Jackson Immunoresearch 115-545-003), goat anti-rabbit Rhodamine Red (Jackson ImmunoResearch, 111-295-003), goat anti-mouse Rhodamine Red (1:200, Jackson ImmunoResearch, 115-295-003), donkey anti-goat Alexa Fluor594 (1:200,

Invitrogen, A-11058), goat anti-rabbit Alexa Fluor633 (1:200, Invitrogen, A-21070), or donkey anti-mouse Alexa Fluor647 (1:200, Invitrogen, A-31571), in PBS-Tx-1%FBS. Following 10 min incubation with bisbenzimide H33258 (1:10,000 in PBS, Sigma, B2883) for labeling nuclei, sections were washed and mounted on glass slides in Mowiol (17% polyvinyl alcohol 4–88, 33% glycerin and 2% thimerosal in PBS). Antigen retrieval was performed when using the mouse anti-PCP4 antibody (30 min at 90˚C in citrate buffer pH6, before the FBS treatment).

Multichannel fluorescence stacks from recorded cells were acquired with a confocal microscope (Leica SP5) with LAS AF software v2.6.0 build 7266 (Leica) and objectives HC PL APO CS 10.0x0.40 DRY UV, HCX PL APO lambda blue 20.0x0.70 IMM UV and HCX PL APO CS 40.0x1.25 OIL UV. Pinhole was set at 1 Airy, and the following channels settings applied (fluorophore, laser and excitation wavelength, emission spectral filter): (a) bisbenzimide, Diode 405 nm, 415 to 485 nm; (b) Alexa Fluor488, Argon 488 nm, 499 to 535 nm; (c) Rhodamine Red/Alexa Fluor594, DPSS 561nm, 571 to 620 nm; (d) Alexa Fluor633/Alexa Fluor647, HeNe 633 nm, 652 to 738 nm. Optical section intervals were of 5, 2, and 1 μm for 10x, 20x, and 40x objectives, respectively. Brightness and contrast were adjusted with the ImageJ software (NIH Image). For illustration purposes, z-projections (average intensity) were made in some images. Analyses of colocation, cell position, and/or quantifications were always made at one confocal plane.

For illustration purposes, false colors were used. All morphological analyses were performed blindly with respect to electrophysiological data. The distance from the cell soma to the MF limit (taken as 0) or the cell position within SP (the superficial border taken as 0) was measured from confocal images using information from CB and bisbenzimide staining and the ImageJ software (NIH Image) [31]. The proximodistal distance to MF was measured along the linear SP contour and normalized within each region.

To complement immunostaining studies, we used sections from $n = 3$ VenusA-GFP transgenic rats with specific expression in GABAergic interneurons. VGAT–Venus transgenic rats were generated by Drs. Y. Yanagawa, M. Hirabayashi, and Y. Kawaguchi at the National Institute for Physiological Sciences (Okazaki, Japan) using pCS2–Venus provided by Dr. A. Miyawaki. VGAT line progenitors were provided by the National Bioresource Project Rat (Kyoto, Japan) [73]. Their brains were processed as previousy described.

## Data analysis

Passive electrophysiological properties (input resistance and membrane decay) of all neurons recorded intracellularly in vivo were measured using 500 ms currents step in current-clamp mode. RMP and input resistance were estimated by linear regression between baseline potential data and the associated holding current. Intrinsic firing properties, including action potential threshold and AHP, were estimated from the first spike in response to depolarizing current pulses of 0.2 nA amplitude and 500 ms duration. The firing autocorrelogram (2.5 ms bins) was computed using all detected spikes from the cell. A bursting index was defined as the ratio of the number of complex spikes (minimum of 3 spikes <8 ms interspike interval) over the total number of spikes recorded during theta activity.

LFP theta activity was identified from nonoverlapping segments of continuous oscillations in the 4 to 12 Hz band. For theta cycle detection, signals were band-filtered at 4 to 12 Hz (forward-backward-zero-phase FIR filters). Theta phase-locking firing of single cells was measured using the circular mean and from the firing histograms (each theta cycle was divided into 18 bins). Phase locking was quantified using the MVL of phase distribution from 0 to 1 [28].

SWR were identified using the low-pass filtered (<100 Hz) signals from SR to identify sharp waves and bandpass filtered (100 to 600 Hz) signals from SP to identify ripples.

Candidate events were detected by thresholding >2 to 3 SDs. All pairs of detected events were visually confirmed and artifacts were discarded [26]. DS events were detected similar to sharp-waves using LFP signals from the hilus.The SWR and DS ratios were defined as the ratio between the firing rate at the event peak over the basal firing rate.

### Categorical and Spearman's pairwise correlation analyses

Multidimensional classifiers are nominally used to assign property vectors to suggested neuronal groupings, which are already clearly defined in Hippocampome.org. We, therefore, utilized categorical contingency matrices to explore hidden pairwise relationships between in vitro and in vivo properties, such as between membrane biophysics properties and theta phase locking or between biomolecular marker expression and SWR ratio values, as a way of uncovering the efficacy of in vivo properties in identifying neuron types in extracellular recordings. We expanded upon our prior exploration of pairwise connections between 205 categorical in vitro properties [22], also using Barnard's exact test to evaluate 2 × 2 contingency matrices of these properties, which provides one with the greatest statistical power when row and column totals are free to vary [74]. Contingency matrices again were selected for the analysis of correlations between neuron-type properties, since a majority of the properties now collated in Hippocampome.org (206 out of 248) are nominally categorical in nature. To enable a full analysis of all 248 properties, those that are continuous in nature were converted into categories. The in vitro categorical properties evaluated for each neuron type included the main neurotransmitter, either glutamate or GABA; the projecting or local nature of axons and dendrites; whether or not the axons and dendrites overlap in some layer of the hippocampal formation; whether or not the axons or dendrites exist in a single layer; whether or not the axons or dendrites exist in 3 or more layers; the presence or absence of the axons, dendrites, and soma in the 26 layers of the hippocampal formation; whether or not the soma exists solely in the principal cell layer; clear positive or negative expression of any of 98 biomolecular markers; the top third or bottom third of values of 10 membrane biophysics properties; the presence or absence of any of the 24 firing pattern phenotypes found in the hippocampal formation; and high or low values for 4 single-compartment Izhikevich modeling parameters. For the new in vivo properties, we took several approaches to categorize the data. For the theta phase locking, we determined whether the specific phase values aligned with either the peak or the trough of the calibration LFP with its peak occurring at 0˚ in CA1 stratum pyramidale. For the SWR ratio values, we categorized the data as being either less than 1 or greater than or equal to 1. In a manner similar to the treatment of the in vitro membrane biophysics properties, we categorized the baseline in vivo firing rates as being either in the higher one-third or lower one-third of values.

As a check on our categorical analysis with contingency matrices, using the same 248 properties, we also ran an analysis utilizing Spearman's rank correlation coefficient [75]. This time, however, we evaluated discrete ordinal values, such as the presence (1) or absence (0) of neurites in a given layer, and continuous values, such as the actual values for the theta phase locking [0, 360].

### Supporting information

**S1 Fig. Data processing and alignment of LFP signals across hippocampal layers and regions.** (**A**) Method to ascribe a particular cell to a Hippocampome.org neuron type. (**B**) Equivalence between reported theta phases of single cells in a given region/layer and the reference theta clock at CA1. Note our convention of 0˚ at the CA1 SP theta peak. (**C**) Left: Coronal section of the dorsal hippocampus showing the track of the silicon probe shank (discontinuous

line) across layers. Center/right: Representative LFP traces across layers from CA1 (red) to the DG (blue) during theta oscillations and (right) SWR events. The plot shows the polarity reversal of theta cycles across layers for theta oscillations recorded during RUN and REM sleep. CA1 layers: SO, stratum oriens; SP, stratum pyramidale; SR, stratum radiatum; and SLM, stratum lacunosum moleculare. DG layers: ML, molecular layer; GC, granule cell layer; and the hilus. Scale bar 1 mm. Data from this work. (**D**) Left: Section of the EC showing the reconstructed tracks of the four-shank silicon probe (200 μm inter-shank distance). Numbers indicate the EC layers. Scale bar 1 mm. Right: Phase relationship between hippocampal events (SWR and theta cycles) recorded across EC layers. Note the similar phases across layers 5 to 2 and polarity reversal in EC1. Modified from Mizuseki and colleagues [39]. AB, angular bundle; DG, dentate gyrus; EC, entorhinal cortex; GC, granule cell; LFP, local field potential; ML, molecular layer; REM, rapid-eye movement; RUN, running; SWR, sharp-wave ripple. (TIF)

**S2 Fig. Network dynamics of morphologically identified CA3 and CA2 pyramidal cells.** (**A**) Proximal CA3 pyramidal cell recorded from the CA3c region. Note thorny excrescences characteristics of this cell type. Another CA3c cell recorded near the border with CA3b is shown at bottom, and their apical dendrites indicated by arrowheads. The Hippocampome. org morphological encoding is shown at right (purple: axon and dendrites in layer; blue: dendrites in layer). (**B**) Firing properties of the 2 CA3 pyramidal cells shown in A in response to current pulses are consistent with nonadapting firing and silence preceded by transient slow-wave bursting (TSWB.NASP and TSWB.SLN). AP waveform and firing autocorrelogram are shown at right. (**C**) Examples of a proximal CA2 cell (CA2b; top) and a distal CA2 cell (CA2a; bottom). Cell identity is confirmed with immunostaining against CB and PCP4. Note that CA2 pyramidal cells are all negative to CB (Table 3). See also Fernandez-Lamo and colleagues [31] for immunoreactivity against α-actinin2. The Hippocampome.org biomolecular marker encoding is updated to clarify this ambiguity (green-blue triangle: positive–negative unresolved for subtypes; green triangle: positive; blue triangle: negative). (**D**) Firing properties of CA2 cells shown in C. (**E**) Membrane biophysics properties of CA3 and CA2 cells recorded in vivo as compared with in vitro data from Hippocampome.org. (**F**) Firing pattern of CA3 cells during SWR, theta, and DS. (**G**) Firing pattern of CA2 cells during SWR, theta, and DS. (**H**) Group mean theta phase-locking preferences of all CA3 and CA2 pyramidal cells in the database (upper plot). Same for the SWR ratio (middle plot) and for the DS ratio (bottom plot). Data from $n = 7$ CA3 cells and $n = 5$ CA2 pyramidal cells. See Table 2. The underlying data can be found in S1 Data. AP, action potential; CB, calbindin; DS, dentate spikes; HIL, hilus; MF, mossy fiber; ML, molecular layer; NASP, non-adapting spiking; SL, stratum lucidum; SLN, silence; SO, stratum oriens; SP, stratum pyramidale; SR, stratum radiatum; SWR, sharpwave ripple; TSWB, transient slow-wave bursting. (TIF)

**S3 Fig. In vivo network behavior of CA1 pyramidal cells.** (**A**) Example of a superficial pyramidal cell recorded intracellularly from the dorsal CA1 region. Note location within the CB-positive sublayer. Somadendritic reconstruction after Neurobiotin processing is shown at right. (**B**) Example of a CB-negative deep pyramidal cell. The Hippocampome.org morphological and molecular marker encodings are shown at right. Note some unresolved expression data in v1.8 of Hippocampome.org (CA1 pyr cells), which are clarified by introducing the deep and superficial subtypes. Data on differential expression level is based on hipposeq. janelia.org (Cembrowski and colleagues [55]). The blue-green triangles indicate unclear positive–negative evidence for subtypes, red triangles for positive–negative (unresolved), magenta triangles for positive–negative (subcellular expression differences), and green triangles with

pluses for differential expression favoring one subtype over the other. This latter depiction of gradients is for illustrative purposes only, as the molecular marker matrix of Hippocampome. org does not grade positive expression; however, Hippocampome.org reports information on expression gradients both in the evidential pages as well as in the Notes section of the neuron pages (Table 1). (**C**) Firing properties of the superficial CA1 pyramidal cell shown in A in response to current pulses consistent with ASP. The spike waveform and the firing autocorrelogram are also shown**. (D)** Membrane biophysics properties of superficial CA1 cells recorded in vivo as compared with in vitro data from Hippocampome.org. (**E, F**) Same for the deep CA1 pyramidal cell shown in B. (**G**) Firing pattern of the superficial CA1 pyramidal cell shown above during SWR, theta, and DS. (**H**) Same for the deep cell shown above. (**I**) Group mean theta phase-locking preferences of all CA1 pyramidal cells in the database (top plot). Same for the SWR ratio (bottom). Statistical differences between deep and superficial cells are evident (T: Student $t$ test; U: Mann–Whitney test). Data from $n = 11$ deep and $n = 10$ superficial CA1 pyramidal cells. See also Table 2. The underlying data can be found in S1 Data. AP, action potential; ASP, adapting spiking; CB, calbindin; DS, dentate spikes; HIL, hilus; SLM, stratum lacunosum moleculare; SO, stratum oriens; SP, stratum pyramidale; SR, stratum radiatum; SWR, sharp-wave ripple.
(TIF)

**S4 Fig. Intracellular recording of PV+ basket cells.** (**A**) Morphological and neurochemical identification of a PV basket cell recorded in vivo with sharp glass pipettes. (**B**) Firing properties of the PV basket cell shown in A in response to current pulses. The spike waveform and firing autocorrelogram are shown at right**. (C)** Membrane biophysics properties of PV basket cells recorded in vivo as compared with in vitro data from Hippocampome.org. (**D**) Firing pattern of the PV basket cell shown in panels A and B during SWR and theta oscillations. The underlying data can be found in S1 Data. AP, action potential; HIL, hilus; PV, parvalbumin; SO, stratum oriens; SP, stratum pyramidale; SST, somatostatin.
(TIF)

**S1 Table. Formal names and descriptors of neuron types and supertypes in Hippocampome.org.**
(XLSX)

**S2 Table. Compilation of in vivo data mined from literature.**
(XLSX)

**S3 Table. Individual in vivo data values summarized in Tables 1 and 2.**
(XLSX)

**S1 Data. Excel spreadsheet containing, in separate sheets, the underlying numerical data for Figure panels 2 A–E, 3C, 3D, 3F, 3G, 3 I–K, 4 B–F, 5A, 6E, S2E–S2H, S3D, S3G–S3I, S4C, and S4D.**
(XLSX)

## Acknowledgments

We thank György Buzsáki for generous support.

## Author Contributions

**Conceptualization:** Liset M. de la Prida, Giorgio A. Ascoli.

**Data curation:** Alberto Sanchez-Aguilera, Diek W. Wheeler, Teresa Jurado-Parras, Manuel Valero, Elena Cid, Ivan Fernandez-Lamo, Daniel García-Rincón.

**Formal analysis:** Alberto Sanchez-Aguilera, Diek W. Wheeler, Teresa Jurado-Parras, Miriam S. Nokia, Elena Cid, Daniel García-Rincón.

**Funding acquisition:** Liset M. de la Prida, Giorgio A. Ascoli.

**Investigation:** Diek W. Wheeler, Teresa Jurado-Parras, Manuel Valero, Elena Cid, Ivan Fernandez-Lamo, Daniel García-Rincón, Liset M. de la Prida, Giorgio A. Ascoli.

**Methodology:** Liset M. de la Prida, Giorgio A. Ascoli.

**Project administration:** Liset M. de la Prida, Giorgio A. Ascoli.

**Resources:** Diek W. Wheeler, Nate Sutton, Giorgio A. Ascoli.

**Software:** Diek W. Wheeler, Nate Sutton.

**Supervision:** Liset M. de la Prida, Giorgio A. Ascoli.

**Validation:** Diek W. Wheeler, Liset M. de la Prida, Giorgio A. Ascoli.

**Visualization:** Alberto Sanchez-Aguilera, Diek W. Wheeler, Nate Sutton, Liset M. de la Prida, Giorgio A. Ascoli.

**Writing – original draft:** Diek W. Wheeler, Liset M. de la Prida.

**Writing – review & editing:** Alberto Sanchez-Aguilera, Diek W. Wheeler, Teresa Jurado-Parras, Manuel Valero, Miriam S. Nokia, Elena Cid, Ivan Fernandez-Lamo, Nate Sutton, Daniel García-Rincón, Giorgio A. Ascoli.

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
