## [Editor Report · Decision Letter 0]

24 Nov 2020

Dear Dr Wheeler, 

Thank you for submitting your manuscript entitled "Rhythmic firing of hippocampal and entorhinal neuron types in vivo: integrating single-cell phenotypes with circuit function" for consideration as a Methods and Resources article by PLOS Biology.

Your manuscript has now been evaluated by the PLOS Biology editorial staff, as well as by an academic editor with relevant expertise, and I am writing to let you know that we would like to send your submission out for external peer review.

Please re-submit your manuscript within two working days, i.e. by Nov 30 2020 11:59PM.

Kind regards,

Gabriel Gasque, Ph.D.,

Senior Editor

PLOS Biology

---

## [Decision Letter · Decision Letter 1]

25 Jan 2021

Dear Dr Wheeler,

Thank you very much for submitting your manuscript "Rhythmic firing of hippocampal and entorhinal neuron types in vivo: integrating single-cell phenotypes with circuit function" for consideration as a Methods and Resources Article at PLOS Biology. Your manuscript has been evaluated by the PLOS Biology editors, by an Academic Editor with relevant expertise, and by three independent reviewers. You will note that reviewers 2 and 3, Linda Katona and Szabolcs Káli, respectively, have revealed their identities. 

In light of the reviews (below), we will not be able to accept the current version of the manuscript, but we would welcome re-submission of a much-revised version that takes into account the reviewers' comments. We cannot make any decision about publication until we have seen the revised manuscript and your response to the reviewers' comments. Your revised manuscript is also likely to be sent for further evaluation by the reviewers.

We expect to receive your revised manuscript within 3 months. 

**IMPORTANT - SUBMITTING YOUR REVISION**

Your revisions should address the specific points made by each reviewer. As you will see from their detailed comments, the reviewers are fairly enthusiastic about the improvements you have made to the Hippocampome.org resource. Reviewer 2 does ask for one additional analysis to help clarify how well the updated Resource can distinguish between similar cell types, and reviewer 3 notes a variety of issues that would need to be fixed on the updated Hippocampome.org website. In addition, all the reviewers felt that the study would benefit from some re-writing and reorganization to help clarify the findings.

Editorially, we think the Introduction reads well. However, the use of PoE and PoK abbreviations for “pieces of evidence” and “pieces of knowledge” is distracting and creates confusion in the Results. More importantly, the Results section on the curated data reads more like a Methods section of how you did the curation, rather than providing clear insights into what you found from curating these data. While this likely reflects that you found a lot of data, and used it to update the Hippocampome.org website, these sections of the Results would be stronger if you could specifically provide a table and write-up of all the key findings in a systematic way (rather than describing a series of examples in a narrative). The later part of the Results is better, providing some useful analyses of how this expanded website can be useful to the community. 

Please submit the following files along with your revised manuscript:

*Re-submission Checklist*

*Published Peer Review*

*PLOS Data Policy*

*Blot and Gel Data Policy*

Sincerely,

Gabriel Gasque, Ph.D.,

Senior Editor,

ggasque@plos.org,

PLOS Biology

REVIEWS:

Reviewer #1: This manuscript by Sanchez-Aguilera and colleagues, reports an update on the authors' ongoing efforts to develop and curate an online database (Hippocampome.org) that contains a comprehensive catalogue of hippocampal and entorhinal cortical cell types with available literature information about the firing dynamics during in vivo oscillations. Furthermore, the authors extend this knowledge with new data from in vivo recordings from hippocampal principal cells and GABAgergic interneurons as well as with a reanalysis of data from published CA2 and CA1 pyramidal cells. 

I commend and congratulate the authors for this efforts here. The rodent hippocampal circuitry remains one of the most intensely investigated cortical circuit with ever increasing data on its cell types. Despite this premise, to fully leverage this immense knowledge, it requires efforts like this one presented here to catalogue and a synthetize the fragmented information across different labs and experimental conditions. In this respect the authors ongoing efforts are absolutely critical for a better understating of the relationship between cellular and circuit architecture and physiological, and perhaps ultimately behavioral, functions of the hippocampal-entorhinal circuitry.

I think this update on the authors' continued efforts would warrant a publication. Obviously, there are many other exciting potential directions this database can be further developed by for example integrating with available information on connectivity and genetic information on hippocampal cells types, which would make the effort perhaps even more compelling to a general audience, however it is arguably beyond the scope of the current iteration of the database update. 

Reviewer #2, Linda Katona: In this manuscript the authors present an open-science classification system they developed which enables the integration of single-cell phenotypes with the functional understanding of their complex networks. This is a very timely update to an existing online dataset (Hippocampome.org) of hippocampal and entorhinal cortical neuron types. The major breakthrough arises from data mining the current literature and mapping the in vivo oscillatory-state dependent activity patterns of single cells to Hippocampome.org cell types. Furthermore, it is very important that the authors complement gaps in our knowledge system by recording underrepresented and missing cell types under different conditions. The conclusion is that such knowledge-based explorations can promote augmented understanding of hippocampal-entorhinal function.

This is a much awaited common framework which should permit the annotation of genetically targeted high-throughput recordings based on the much fewer but morphologically and neurochemically identified neuron types. Overall the manuscript is impressive but there are some major points that need to be addressed before it can meet publication criteria. I have also included additional minor comments to improve the presentation.

Major comments

1.The authors demonstrate the system's successful application. Would the authors expect for it to perform just as well if the data are more similar i.e. SST+ O-LM cells vs SST+ bistratified cells vs SST+ long-range projecting GABAergic neurons? Could this be tested with some existing dataset? Have the authors considered including oscillatory state-dependent action potential autocorrelograms as categorical variables to improve on the classification?

2a.The authors identified gaps in the knowledge-base and have chosen appropriate recording techniques to acquire the required data to fill these gaps. However, more detail should be provided in the methods section about the analytical tools used (i.e. pairwise correlation) and please justify your choice in comparison to other multidimensional classifiers. 

2b.The authors should explain the categorical properties listed, especially those related to the newly introduced firing pattern phenotypes (page 32, 2nd para, Methods, Pairwise correlation analysis).

3.The discussion is quite lengthy and speculative in places (e.g. 2nd para on page 25). Please make it more concise.

4.I could not find this relevant publication listed in Supp. Table 2: Extended interneuronal network of the DG (Szabo et al., 2017 Cell Reports). If not there yet, it should be uploaded to the Hippocampome.org as it complements nicely the freshly acquired data.

Minor points

1.page 4, 1st para, Results, Mapping single-cell oscillatory dynamics to Hippocampome.org cell types 

Please update your figure referencing, it should not start with a Supplementary Figure and Table in the main text. 

2.page 31, 1st para, Methods, Histology and immunostaining 

Multichannel fluorescence stacks were acquired... Please give details about the objectives used, the spectral channel separation (beam splitter settings), pin hole size, optical section thickness and Z stack height. Was there any post processing applied to the images (filtering, Z-projections)? 

3.Missed citations/referencing: 

-page 3: Single-cell phenotyping based on firing behavior in vivo has provided a large body of evidence to characterize microcircuit operation [7-10, 75, 21, 25].

-page 4: By integrating in vivo functional PoKs (e.g. preferred theta phases and SWR-associated firing dynamics) into Hippocampome.org, we extended the previous classification of certain neuron types using novel information on oscillatory dynamics [19].

-page 5: to the Collateral-related cell superfamily (Supp. Table 1)

-page 5: Neurons morphologically identified and/or tested against a battery of neurochemical markers enabled their assignment to specific Hippocampome.org types (Fig. 1)

-page 7: For instance, GABAergic CA1 bistratified cells preferentially fire during the ascending ripple phase (Fig.1A, [21]), while O-LM cells tend to decrease their firing rate during SWR as compared with baseline (Fig.1B, [7]); but see also [25, 7] for heterogeneous participation of these cells during a variety of SWR.

-page 8: most CA1 pyramidal cells fire near the theta trough (Fig.1D, [ref?]). Finally, single-cell firing dynamics during gamma (25-90 Hz) and/or so-called epsilon (90-130 Hz) oscillations were reported in a minority of cases (Fig.1E) [ref?]

Reviewer #3, Szabolcs Káli: The manuscript describes a rather complex body of work whose primary purpose is to provide curated and consolidated information about the firing properties of hippocampal and entorhinal cortical neurons in relation to the main hippocampal population activity patterns. This information is important from several different aspects: it allows the identification of cell types from extracellular recordings in vivo (as demonstrated by the authors), defines fundamental constraints for the validation of realistic network models, and provides a conceptual framework for theories and further experimental studies of hippocampal network function. Results presented in this paper include the curation of data from the experimental literature, which was a highly nontrivial task in itself despite the relatively limited number of relevant articles, due to the large heterogeneity of experimental preparations and recording methods; new as well as previously published but reanalyzed experimental data from several different areas, cell types, and conditions; and, importantly, the integration of this new knowledge into the online database Hippocampome.org.

While the overall aim of the study is clearly of high importance, the execution of the work (especially the experimental part) is of high quality, and most details are presented clearly, the coherence of the manuscript could be improved substantially. I sometimes had the impression of reading two separate, but interleaved papers - one about how information about the firing of hippocampal neurons relative to network events was collected and used, and another one about the various fascinating but sometimes loosely related findings from a variety of new experiments in vivo. In my view, this issue stems from the dilemma that, on the one hand, many of the new experimental findings would deserve a more thorough description and discussion but, on the other hand, some of the existing discussion already distracts the reader from what is declared to be the main topic of this paper.

I commend the authors for making available for review a new version of Hippocampome.org that includes the new data and other improvements associated with this paper. I spent some time exploring this new version of the website, and I would like to mention some issues that the authors may want to address before the paper is published and the new version becomes public.

First, I was unable to locate some of the new data from the current manuscript on Hippocampome.org. In particular, it seems to me that the data on in vivo membrane properties (presented in Table 1 of the paper) has not been added to the appropriate part of Hippocampome.org.

Second, while the creation of new subcategories for certain cell types (e.g., deep and superficial CA1 pyramidal neurons) is certainly justified by the available data (both in the current manuscript and other recent papers), this creates various challenges for the database and the user interface, some of which appear to be currently unsolved. For instance, the various tables of cellular properties (e.g., the "electrophysiology matrix") now list CA1 Superficial PCs and CA1 Deep PCs instead of CA1 PCs, but the table entries (and associated references) for both new cell types were apparently copied over from the earlier CA1 PC entry, and therefore give identical values for the two new subtypes. This gives the false impression that (at least according to our current knowledge) these two cell types have identical intrinsic physiological properties, even though the current manuscript (as well as other papers) now provide ample evidence to the contrary. In my opinion, the creation of new subtypes should be treated consistently in every part of the database - although I assume that this is a difficult task which may require manual re-evaluation of relevant parts of the literature. In addition, a way should be found to include and clearly display data that are specifically about the newly created subtypes, but also data which are associated with the more general cell type (i.e., where the subtype is unknown).

Third, the ability to filter data points according to a variety of factors is a great new feature which considerably enhances the usability of the database, especially given the substantial differences that are sometimes observed between experimental conditions. However, this feature does not currently seem to work as expected. For example, if I select a combination of factors (conditions) for which there are currently no data available (or if I simply deselect all factors) and press the "update" button, the table gets updated, but there are still values shown in every cell of the table (except in a few that are always empty) - although when I hover the cursor over these entries, the system (correctly) shows the number of relevant sources and measurements as 0. I also suggest that, when this feature works correctly, the functionality should eventually be extended to all parts of the database (although this work is clearly beyond the scope of the current manuscript, and should not affect its publication). Similarly, I found the graphical representation of the data in Figure 5A to be very informative and easier to digest than the table format in Figure 5B that is currently included in Hippocampome.org, so it may be worth considering the addition of such as graphical representation (possibly in an interactive format) to the website.

Although the reasoning in the paper is generally quite sound, I cannot agree with the conclusions of the paragraph in lines 410-416. Here, after observing that the newly added in vivo firing properties of neurons do not show significant pairwise correlations with any of the previously described (in vitro) characteristics, the authors conclude that "these in vivo properties are largely independent and could thus add classificatory power when combined with traditional in vitro characterization." However, the fact that individual (linear) correlations with individual variables are not statistically significant does not imply that the new features could not be predicted (possibly with high confidence) by some (potentially nonlinear) combination of all the in vitro features. In fact, it would be surprising if the new features and the older ones were truly independent as both of these are shown to be strongly dependent on the cell type. Combinations of morphological, electrophysiological, and molecular features have proved to be good predictors of cell type (as demonstrated beautifully by Hippocampome.org), and the cell type is a good predictor of in vivo firing behavior (as shown by the data summarized in the current manuscript). On the other hand, I agree with the claim that the newly added features may be uniquely useful for the categorization of neurons in the context of in vivo ensemble recordings where the majority of the other features that are traditionally used for classification are typically not available.

Finally, I suggest that the manuscript may benefit from some thorough proofreading as I found several typos as well as minor errors in grammar and usage throughout the text.

Minor issues:

- I think that intracellular recordings using sharp micropipettes are more commonly referred to as "sharp electrode recordings" rather than "sharp recordings"

- the reference "Rees et al., 2017" on line 467 appears to be incorrectly formatted, and is not included in this form in the list of references (although there is an entry for Rees et al., 2016)

---

## [Editor Report · Decision Letter 2]

22 Mar 2021

Dear Dr Wheeler,

Thank you for submitting your revised Research Article entitled "Rhythmic firing of hippocampal and entorhinal neurons in vivo: integrating single-cell phenotypes with circuit function" for publication in PLOS Biology. I have now discussed your revision with the editorial team and with the Academic Editor as well. 

I am pleased to let you know that we are editorially satisfied with your manuscript. However, before I can move ahead with your paper and send it to our production team for final and formal acceptance, we need to make sure you address the following data and other policy-related requests:

**Title:

We suggest a new title that is less descriptive and more instructive, and that mentions Hippocampome.org: "An update to Hippocampome.org by integrating single-cell phenotypes with circuit function in vivo." If you think this title misrepresents your work, let me know and we can work together on an alternative.

**Ethics:

-- Please include ID approval number for your protocol(s) approved by the Institutional Animal Care and Use Committee (IACUC) at New York University Medical Center.

**Data:

We note that much of your data can be found at http://www.hippocampome.org. However, we also ask for all individual quantitative observations that underlie the data summarized in the figures and results of your paper. For an example see here: http://www.plosbiology.org/article/info%3Adoi%2F10.1371%2Fjournal.pbio.1001908#s5

These data can be made available in one of the following forms:

Regardless of the method selected, please ensure that you provide the individual numerical values that underlie the summary data displayed in the following figure panels: Figures 2A-E, 3CDFGIJK, 4B-F, 5A, 6E, S2EFGH, S3DFGHI, and S4CD.

Please also ensure that each figure legend in your manuscript includes information on where the underlying data can be found

Please ensure that your supplemental data file/s has/have a legend.

We expect to receive your revised manuscript within two weeks. 

*Published Peer Review History*

*Early Version*

Sincerely,

Gabriel Gasque, Ph.D.,

Senior Editor,

ggasque@plos.org,

PLOS Biology

DATA NOT SHOWN?

---

## [Editor Report · Decision Letter 3]

30 Mar 2021

Dear Dr Wheeler,

On behalf of my colleagues and the Academic Editor, Thomas Klausberger, I am pleased to say that we can in principle offer to publish your Research Article "An update to Hippocampome.org by integrating single-cell phenotypes with circuit function in vivo" in PLOS Biology, provided you address any remaining formatting and reporting issues. These will be detailed in an email that will follow this letter and that you will usually receive within 2-3 business days, during which time no action is required from you. Please note that we will not be able to formally accept your manuscript and schedule it for publication until you have made the required changes.

In addition, please make the following two changes to your manuscript:

1) Please ensure that your Supporting File S1 Data has a legend

2) Please include the ID number for the protocol(s) approved by the Ethics Committee of the Instituto Cajal (CSIC). (Please accept my apologies since I should have requested this in my previous decision letter).

PRESS

Thank you again for supporting Open Access publishing. We look forward to publishing your paper in PLOS Biology. 

Sincerely, 

Gabriel Gasque, Ph.D. 

Senior Editor 

PLOS Biology